

# Quantitative Analysis of Nighttime Effects of Radiation Belt Energetic Electron Precipitation on the D-Region Ionosphere during Lower Solar Activity Period

Xuan Dong[123], Shufan Zhao[123*], Li Liao[4*], Wei Xu[5], Ruilin Lin[123], Xiaojing Sun[6], Shengyang Huang[123],

Yatong Cui[12], Jinlei Li[127], Hengxin Lu[8], Xuhui Shen[123*]

[1]State Key Laboratory of Space Weather, National Space Science Center, CAS, Beijing, 100190, China

[2]Key Laboratory of Solar Activity and Space Weather, National Space Science Center, CAS, Beijing, 100190, China

[3]University of Chinese Academy of Sciences, Beijing, 100190, China

[4]Institute of Geophysics, China Earthquake Administration, Beijing, 100081, China

[5]Department of Space Physics, School of Electronic Information, Wuhan University, Wuhan, 430072, China

[6]BeiHang University, Beijing, 100191, China

[7]The China University of Geosciences (Beijing), Beijing, 100083, China

[8]National Institute of Natural Hazards, Ministry of Emergency Management of China, Beijing, 100085, China

*Correspondence to*: Shufan Zhao (zhaoshufan@nssc.ac.cn), Li Liao (liaoli@cea-igp.ac.cn), Xuhui Shen

(shenxuhui@nssc.ac.cn)

**Abstract.** Energetic electron precipitation (EEP) from the Earth's radiation belts can ionize neutral molecules in the D-region ionosphere (60–90 km altitude), significantly influencing the conductivity and chemical species therein. However, due to the limited resolution of space-borne instruments, the energy and fluxes of electrons that truly precipitate into the atmosphere

remain poorly investigated. To resolve this problem, in this study, we have utilized the wave and particle data measured by the Electric Field Detector (EFD) and High-Energy Particle Detector (HEPP) onboard the CSES-01 satellite during nighttime conditions between 2019 and 2021. Using the measurements of Extreme-Low-Frequency (ELF) waves, we have derived the reflection height of the D-region ionosphere, which turn out to be highly consistent with the electron and X-ray measurements of CSES. Our results show that the influence of EEP on the two hemispheres is asymmetric: the reflection height in the

Northern Hemisphere is in general lowered by 2.5 km, while the reflection height in the Southern Hemisphere is lowered by 1.5 km, both of which are consistent with first-principles chemical simulations. We have also found that the decrease of reflection height exhibits strong seasonal variation, which appears to be stronger during winter time, and relatively weaker during summer time. This seasonal difference is likely related to the variation of the background ionospheric electron density. Our findings provide a quantitative understanding of how EEP influences the lower ionosphere during solar minimum periods,

which is critical for understanding the magnetosphere-ionosphere coupling and assessing the impact on radio wave propagation.



## 1 Introduction

The D-region ionosphere (60–90 km) is a complex and dynamic medium composed of electrically charged and neutral particles. It is influenced by solar activity and the precipitation of energetic particles from the magnetosphere (Kumar and Kumar, 2018). Energetic electron precipitation (EEP) from the Earth's radiation belts plays a significant role in altering the ionosphere's composition and dynamics (Blake et al., 1996; Nakamura et al., 1995; Matthes et al., 2017; Seppälä et al., 2014). High-energy particles, particularly electrons with energies ranging from 100 keV to several MeV, can penetrate deeply into the D-region ionosphere. The main types of collisional processes between EEP and neutral molecules include elastic scattering, ionization process, and bremsstrahlung radiation, which generate X-rays. The ionization process produces electron-ion pairs that affect the chemistry of both the D-region ionosphere and the neutral atmosphere (Krause, 1998; Randall et al., 2005; Randall et al., 2007; Pettit et al., 2023). The ionization caused by these energetic particles can significantly change the electron density and impact the reflectivity of radio waves in this region, such as Extremely-Low-Frequency (ELF, 30 Hz to 3 kHz) and Very-Low-Frequency (VLF, 3 kHz to 30 kHz) waves (Cai et al., 2023; Ma et al., 2024; Zhao et al., 2024; Zhao et al., 2019; Yang et al., 2018).

EEP is typically driven by wave-particle interactions, such as cyclotron and Landau resonances. These mechanisms scatter trapped electrons into the loss cone, enabling them to descend to low altitudes (around 100 km) and deposit their energy through collisions with atmospheric gas molecules (Kataoka et al., 2020; Li et al., 2019; Ma et al., 2020; Ma et al., 2021). Previous studies have concentrated on the atmospheric response to precipitating particles. When these high-energy electrons interact with the neutral atmosphere, they produce complex effects in both the ionosphere and the atmosphere, including changes in electron density and the generation of NOx and HOx species (Andersson et al., 2012; Arsenović et al., 2016; Funke et al., 2011; Fytterer et al., 2015; Jackman et al., 2001; Verronen et al., 2011a; Verronen et al., 2011b). These changes alter the ionosphere's characteristics and can significantly impact the ozone layer and overall atmospheric composition (Andersson et al., 2014; Randall et al., 2007; Rozanov et al., 2012; Seppälä et al., 2015; Sinnhuber et al., 2012; Thorne, 1980).

The influence of EEP on the lower ionosphere, particularly the D-region as the primary region of EEP, is of critical importance and has been extensively studied. However, directly studying its effects and performing quantitative analyses remain challenging. On the one hand, it is because the physical, chemical, empirical, and theoretical models (Burns et al., 1991; Friedrich et al., 2018; Verronen et al., 2016) only represent a climatological mean of the D region (Renkwitz et al., 2023); On the other hand, directly studying its effects remains challenging due to limited observational capabilities at these altitudes. This range is too low for space-borne instruments and too high for balloon-based instruments. High-power incoherent scatter radars (ISRs) require long integration times and strong ionization rates, and they are only available in limited locations. Using the Wait and Spies (WS) formula (Wait, 1964), researchers can parameterize the D-region ionosphere. Combined with VLF remote sensing technology, this allows observation of disturbances caused by EEP between VLF transmitters and receivers (Cummer



et al., 1997; Kulkarni et al., 2008). However, the spatial coverage of ground-based observations is limited. The method
proposed by Toledo-Redondo et al. (2012) derives the reflection height by measuring the cutoff frequency of ELF waves
detected by satellites, enabling the global distribution of low ionospheric reflection heights to be measured. This technology
allows us to utilize satellite electromagnetic observation data to analyze the global climatological characteristics of the D
region ionosphere influenced by EEP. Recently, Chen et al. used a model to simulate the changes in lower ionospheric electron
density caused by EEP and their impact on TEC. This is an important attempt to quantify the effect of EEP on the D-region
ionosphere. However, their research results were not compared with observational data (Chen et al., 2023). Renkwitz et
al.,(2023) also present local noon climatologies of electron densities in the D region (50–90 km) over northern Norway as
observed by an active radar experiment. Their results show that EEP has a more significant effect on the ionospheric D region
during the winter months. However, the study on the global climatological characteristics of EEP effects is insufficient.

Despite these advancements, previous studies have not comprehensively analyzed the quantitative (refined) effects of EEP on
the D-region ionosphere. Understanding the refined impact of EEP on the D-region ionosphere remains crucial, particularly in
high-latitude regions where EEP is more frequent, as this is essential for accurately assessing its role in the dynamic changes
of the ionosphere. In this paper, we present the first quantitative study of the impact of EEP on lower ionospheric electron
density using data from the CSES-01 satellite and a first-principles-based multi-component chemical model. On this basis, we
further investigated the climatological characteristics of EEP effects and inferred that the seasonal differences might be closely
related to variations in the background electron density. Section 2 outlines the data sets, models, and inversion methods used
in this study. In Section 3, the first part utilizes multi-payload observations from the CSES satellite to reveal the physical
processes involved in the atmospheric response to EEP; the second part provides a quantitative analysis of global precipitation
effects throughout the year; the third part describes the seasonal variability of reflection height in different hemispheres.

## 2 Data and Model Description

### 2.1 CSES Satellite Data

The China Seismo-Electromagnetic Satellite (CSES) is a low-Earth orbit satellite launched in February 2018. It maintains an
orbital altitude of around 507 km and an inclination of 97.4°, covering geographic latitudes up to 65° north to south (Shen et
al., 2018). As a sun-synchronous satellite, CSES's local time (LT) at the ascending and descending nodes is always 02:00 LT
and 14:00 LT, respectively, with a revisit period of five days. Eight scientific payloads are onboard CSES, including the
Search-Coil Magnetometer (SCM), Electric Field Detector (EFD), High-Precision Magnetometer (HPM), GNSS Occultation
Receiver (GOR), Plasma Analyzer Package (PAP), Langmuir Probe (LAP), High-Energy Particle Package (HEPP), and Tri-
Band Beacon Transmitter (TBB). The EFD measures in-situ electric potentials using four spherical aluminum electrodes with
a diameter of 60 mm. The spatial electric field vector is obtained by dividing the voltage difference by the appropriate
separation distances between each pair of spheres (Huang et al., 2018). EFD data span four frequency bands: Ultra Low





Frequency (DC to 16 Hz), Extremely Low Frequency (6 Hz to 2.2 kHz), Very Low Frequency (1.8 to 20 kHz), and High Frequency (18 kHz to 3.5 MHz). This study uses power spectral density data from the ELF band to precisely identify the cutoff frequency point near 1400 Hz.

The High-Energy Particle Package (HEPP) onboard CSES is essential for studying the pitch angle scattering and precipitation of high-energy particles in near-Earth space. HEPP consists of a high-energy band probe (HEPP-H), a low-energy band probe (HEPP-L), and an X-ray monitor (HEPP-X). HEPP-L measures electron fluxes in the energy range from 100 keV to 3 MeV, which are divided into 256 energy channels, each covering 11 keV. The maximum field of view of HEPP-L is $100° × 30°$, composed of nine silicon slice detector units. These units are divided into two groups based on their field of view: five units

with a narrow half-angle of 6.5° and four units with a wide half-angle of 15°. HEPP-X can provide the counts and energy spectra for X-rays with energies between 0.9–35 keV. This detector has a dead time of less than 10 μs, allowing it to measure up to tens of thousands of counts per second without spectral deviation. The electrons monitored by HEPP-L can precipitate into the D-region ionosphere, colliding with atmospheric molecules and altering particle density at those altitudes. In this work, we use HEPP-L data and the ionization chemistry model PyGPI5 to simulate the resulting changes in the D-region ionosphere.

We also use HEPP-X measurements to estimate the areal extent of these ionization patches.

### 2.2 PyGPI5 Model Simulation

    The PyGPI5 model has been employed to simulate electron density variations in the D-region ionosphere caused by high-energy particle precipitation (Kaeppler et al., 2022). The PyGPI5 model consists of two main classes: Ionization and Chemistry.

The Ionization class utilizes the models that Fang et al. (2008, 2010) developed to generate altitude profiles of ionization for ions and electrons (Fang et al., 2008; Fang et al., 2010). Fang et al. (2010) developed an atmospheric ionization parameterization model based on first-principle physics by solving the Boltzmann equation, which describes the ionization effect caused by isotropic precipitating monoenergetic electrons. This method can decompose any incident energy spectrum into multiple continuous monoenergetic components, and by calculating and integrating their ionization effects, it allows for

analysis on an energy grid incorporating other datasets (e.g., satellite data). By inputting monoenergetic energy (keV), electron energy flux (erg/cm²/s), geographic coordinates, and the desired altitude range into the Fang2010 model, an altitude ionization profile for a specific location can be generated. The Chemistry class contains the core of the GPI5 model (Glukhov et al., 1992). The GPI5 model describes the ion-chemical reactions in the D-region ionosphere, where the interactions of positive and negative light ions, heavy ions, and electrons are expressed as a system of stiff ordinary differential equations (ODEs).

The model first obtains the electron density and neutral atmospheric density through the IRI model and MSIS model. The steady-state electron density is then solved using the least squares method as the initial electron density profile. After inputting the ionization rate and precipitation duration, the model can produce the electron density profiles for a given location. By



solving these equations, the GPI5 model simulates the evolution of electron density in the D-region ionosphere. After inputting the ionization rate and precipitation duration, the model can produce electron density profiles for a given location.


Before inputting the energetic particle flux measured by the CSES satellite into the PyGPI5 model, it is necessary to calculate the flux of particles that can interact with the atmosphere. We assume that particles that reach 100 km altitude would collide with atmospheric molecules, the so-called reference altitude typically used for the definition of precipitation electrons (Marshall et al., 2019). As in Liouville's theorem, the phase space density of particles is conserved along their trajectories in

the absence of collisions, and the pitch angles and magnetic field intensities follow the following relation:

$$\frac{\sin^2 \alpha_{507km}}{\sin^2 \alpha_{100km}} = \frac{B_{507km}}{B_{100km}}, \tag{1}$$

where $\alpha_{507km}$ is the pitch angle of particles at the satellite altitude of 507 km. $\alpha_{100km}$ is the pitch angle at 100 km. $B_{507km}$ and $B_{100km}$ are the magnetic field intensities at 507 km and 100 km, respectively. We use the International Geomagnetic Reference Field (IGRF-13) model to obtain the magnetic field intensities $B_{507km}$ and $B_{100km}$ at the corresponding altitudes (Alken et al.,

2021a; Alken et al., 2021b). By calculating the loss cone angle at satellite altitudes (507 km, $\alpha_{LC}$), we can determine the electron fluxes that can precipitate into the atmosphere. Particles with pitch angles smaller than $\alpha_{LC}$ are within the loss cone and thus expected to interact with the atmosphere. Based on the pitch angle distribution data from the CSES satellite, we integrated to obtain electron fluxes within this loss cone and derived the energy flux input for the model by multiplying the differential electron flux by the energy interval and electron energy.


## 2.3 Reflection Height Calculation Using ELF Waves recorded from CSES

To determine the cutoff frequency, we have improved the method of Toledo-Redondo et al. (2012), making it applicable to high-latitude regions. ELF waves generated below the D-region ionosphere (e.g., from lightning discharge) can propagate upwards, and there is a cutoff frequency at the division point between the QTM1 and QTEM propagation modes, approximately

around 1.6–1.8 kHz. Since losses in the Earth-ionosphere waveguide are maximized at the cutoff frequency, there is a minimum in the satellite electromagnetic spectra corresponds to the cutoff frequency. The cutoff frequency carries information about the reflection height of the Earth's ionospheric waveguide. Using the equation $f_1 = c/h'$, where $c$ is the speed of light, the reflection height of the D-region ionosphere can be calculated from the cutoff frequency. The reflection height is the altitude at which electromagnetic waves are reflected within the ionospheric waveguide, primarily determined by electron density. An

increase in electron density lowers the reflection height. This method is suitable for analyzing reflection height over long time scales but is less effective for individual events, as it relies on the average cutoff frequency from numerous lightning events to capture the long-term characteristics of the ionosphere.



In the magnetosphere, auroral hiss, chorus waves, and lower hybrid electrostatic noise also exist, and their downward propagation can influence the determination of the cutoff frequency. As described by Toledo-Redondo et al. (2012), averaging the spectra within a grid to identify the cutoff frequency can be problematic in the high-latitude region, as a single disturbed wave may result in the absence of extrema across the entire grid (shown in Fig. 3, Fig. 4, Fig. 5 of Toledo-Redondo et al. 2012). Therefore, the cutoff frequencies identified in high-latitude regions may be inaccurate. To address this, we first filter out disturbed spectra with multiple extrema, and then average the filtered spectra, effectively eliminating the interference caused

by downward-propagating waves above the satellite. This approach extends the cutoff frequency screening method to high-latitude regions

Based on the analysis of the spectral data obtained from the ELF band of the CSES satellite's EFD payload at a specific moment, the spectral processing involves detrending and smoothing in two steps to obtain the cutoff frequency. First, the 'convolve'

function is used for initial smoothing. The moving average method effectively reduces high-frequency random noise in the data while preserving the main trend of the signal. The advantage of using 'convolve' lies in its straightforward implementation, efficient computation, and flexible window size setting, making it suitable for removing high-frequency noise. Second, the fast Fourier transform convolution function 'fftconvolve' is applied to the initially smoothed data for further processing, enhancing the smoothing effect and further suppressing noise. Compared to direct convolution, 'fftconvolve' is more computationally

efficient, especially for handling large-scale data, and ensures precise retention of signal trends in multi-level smoothing processes. After these two steps, the trend line of the ELF wave is extracted. Subsequently, within the 1400–2000 Hz frequency range, the smoothed spectral data's local minima are identified. A local minimum is selected only if it satisfies the following conditions: it is the smallest value within the range, and the data to its left shows a decreasing trend, while the data to its right shows an increasing trend. Finally, the frequency corresponding to the selected local minimum is determined as the first cutoff

frequency (shown in Fig. 1).



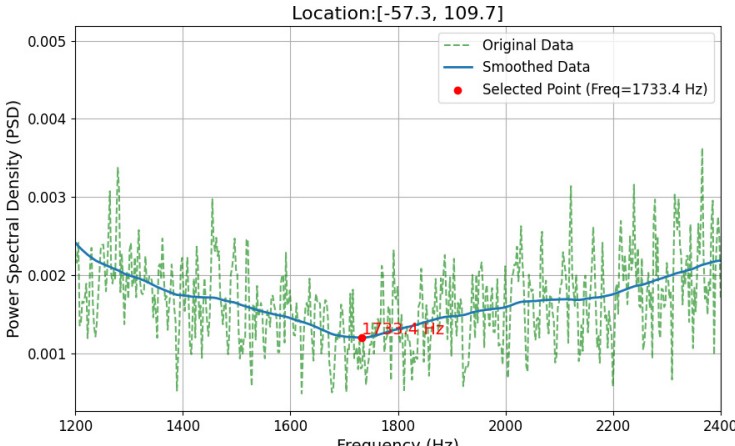

**Figure 1: Extraction of the spectral trend line and identification of the first cutoff frequency in the ELF band from the CSES satellite EFD payload. The green dashed line represents the original spectral data, the blue solid line is the extracted trend line, and the red dots indicate the cutoff frequency corresponding to the local minima.**

**2.4 Reflection Height Calculation Using PyGPI5 Model**

Energetic electrons continuously precipitate into the D-region ionosphere, approaching a steady state as ionized and recombined electrons reach equilibrium. The ionization process of neutral components by precipitating electrons with different energies stabilizes within 10 minutes, while the recombination process of the ionized electrons takes several hours. Consequently, the electron density profile remains consistent from 10 minutes to several hours after the ionization process

occurs. We input the energy spectrum and electron energy flux within the loss cone into the PyGPI5 model, integrating over a 1-hour period to simulate the electron density profile (Shown in Fig. 2).



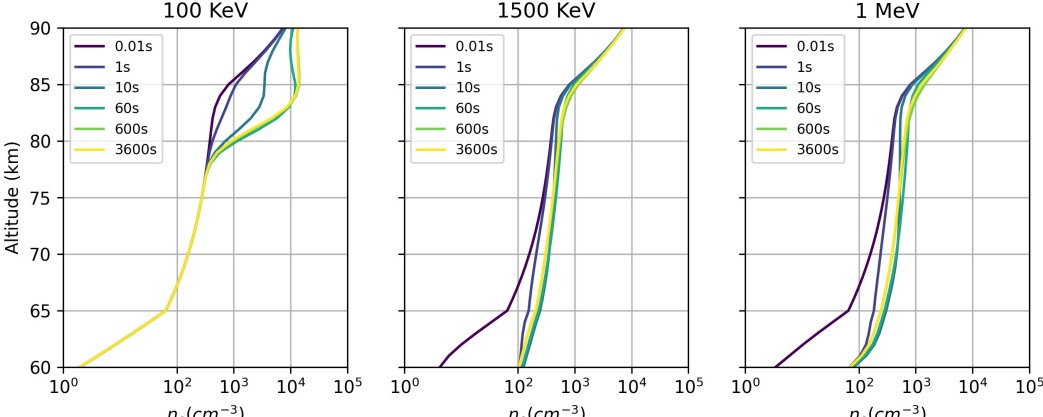

**Figure 2: The electron density at different altitude is plotted to show the relaxation time for a monoenergetic flux at 100 KeV, 1500 KeV, and 1 MeV shown as the left, middle, and right columns, respectively. All examples have an energy flux of 0.01 mW/m².**


To analyze the variation in reflection height from the electron density obtained through the PyGPI5 model, this study employs two methods to calculate the reflection height. One approach (method 1) is to apply the Wait and Spies (WS) formula to model the electron density in the D-region. Another method (method 2) involves utilizing the reflection properties of waves at different frequencies within the ionosphere to determine the reflection heights of these waves. In method 1, We fit the electron

density profile using the WS formula with a residual minimization approach, expressed as follows:

$$N_e(h) = 1.43 \times 10^{13} e^{(-0.15h')} e^{(\beta - 0.15)(h - h')}, \tag{2}$$

where $N_e(h)$ is the electron density at altitude $h$, $h'$ is the reflection height, $\beta$ is the exponential sharpness factor of the ionosphere. The electron density profile is obtained from the PyGPI5 model and fitted using this formula. During the fitting process, we employed Python's curve_fit function to fit the parameters in the formula, including the $\beta$. The curve_fit function

uses a nonlinear least squares approach to iteratively adjust the parameters, ensuring the best fit between the derived curve and the electron density profile calculated by the model. Specifically, curve_fit minimizes the residuals (the differences between the fitted values and the actual values) to optimize $\beta$ and $h'$. This approach ensures both the accuracy and robustness of the fitting process. This fitting allows us to determine the $h'$, enabling a comparison with the reflection height obtained from CSES EFD data through cutoff frequency inversion.

In Method 2, we leverage the physical relationship among the D-region electron density, the effective reflection height $h'$, and the wave frequency. Following Ratcliffe's equation (Ratcliffe, 1959), for cold plasma conditions, the reflection of



electromagnetic waves in the D region occurs when the wave frequency (corresponding to first cutoff frequency $f_1$. in the satellite ELF spectrum) is equal to the plasma frequency squared divided by the collision frequency

$$f_p^2 = f_1 v,\qquad(3)$$

Where $f_p$ is the plasma frequency, $f_1$ is the first cutoff frequency of the waveguide, and $v$ is the collision frequency of the plasma, which can be approximated by an exponential function of altitude. A commonly used model for the collision frequency $v$ at reflection height $h'$ is:

$$v(z) = v_0 e^{-0.15z},\qquad(4)$$

Where $v_0 = 1.86 \times 10^{11} s^{-1}$, and z is the altitude in kilometers. Meanwhile, the plasma frequency $f_p$ in a cold plasma is

related to the electron density $N_e$ by:

$$f_p = 8980\sqrt{N_e},\qquad(5)$$

With $f_p$ in Hz and $N_e$ in cm-3. Combining these relationships (Eq. 3, Eq. 4, Eq. 5), we can calculate the cutoff frequencies of the waves at different heights using Eq. 6.

$$f_1 = 4.34 \times 10^{-4} N_e / e^{-0.15z}\qquad(6)$$

Here, $N_e$ is the corresponding electron density (in cm-3), which can be obtained from the PyGPI5 simulations, $f_1$ is the cutoff frequency calculated from modelled electron density profile. derived from observations, $h''$ is the reflection height (in km). By matching the calculated $f_1$ to the measured $f_1$ (e.g., from CSES EFD data), we can deduce the reflection height $h'$.

We simulated the electron injection of different monoenergetic beams and obtained electron density profiles approaching a

steady state. Additionally, we calculated the reflection height using two different methods, as shown in Fig. 3. Figures 3 (a-d) correspond to cases with no energy injection, 100 keV, 1500 keV, and 1 MeV electron injections, respectively. The results indicate that after electron energy injection, the electron density undergoes noticeable changes. The altitude affected by the electrons varies with energy, with higher-energy electrons penetrating deeper into the atmosphere. Furthermore, we calculated the reflection height using two methods mentioned above. Notably, for the 100 keV electron injection, the reflection height

obtained using Method 2 did not change. This is because 100 keV electrons do not penetrate deep enough to affect the altitude where wave reflection occurs, resulting in no change in the reflection height. However, since the WS profile characterizes the overall changes in the D-region of the ionosphere, the reflection height obtained from WS fitting still shows variations for the 100 keV electron injection. These results confirm that energetic electron precipitation can notably modify the D-region electron density and, consequently, the reflection height. The two methods—WS fitting (Method 1) and Ratcliffe's relation (Method

2)—both capture these variations, though their sensitivities at certain energies may differ.





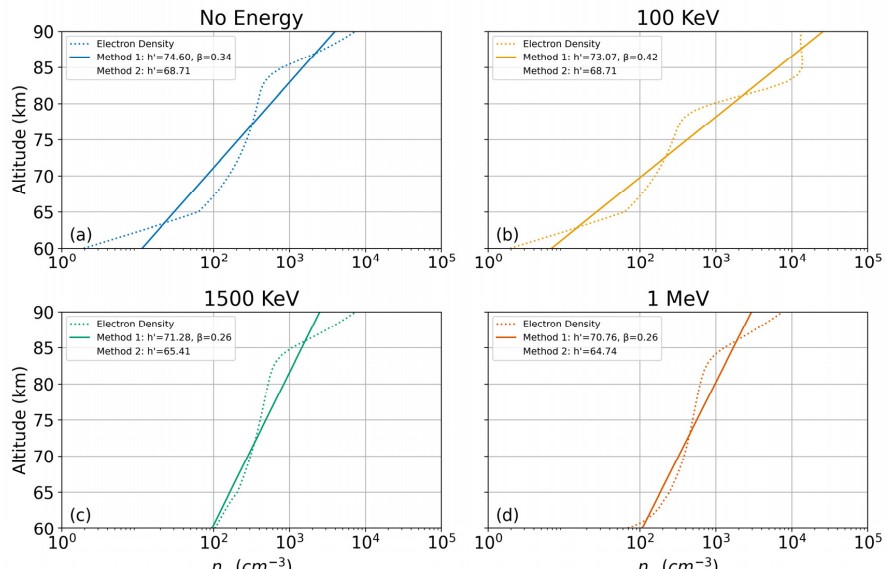

**Figure 3: The fitting process of the WS formula to the electron density curves using the curve_fit function is shown, illustrating the electron density profiles 3600 seconds after the injection of a monoenergetic flux at different energy levels: no energy injection, 100 keV, 1500 keV, and 1 MeV. Except for the no-energy injection case, all examples have an energy flux of 0.01 mW/m². The dashed lines represent the simulated electron density curves, while the solid lines indicate the fitted curves using WS formula. Additionally, the legend also displays the reflection height derived using the method 2, with the cutoff frequency set at 1700 Hz.**

### 3. Results

In 2019–2021, solar activity was at a relatively low level. During the daytime, solar radiation dominated the ionization of atmospheric molecules, causing the reflection height to drop to approximately 70 km. In addition, the elevated electron density in the ionosphere during the day significantly absorbs and attenuates the energy of upward-propagating ELF waves, making it difficult to observe the cutoff frequency point $f_1$. To minimize the influence of solar activity on the observations and eliminate the interference from daytime ionospheric absorption effects, in this study, we only use CSES EFD data collected during nighttime conditions (02:00 LT) to invert the reflection height.

### 3.1 Relationship Between Electron Flux, X-ray Rate, and Reflection Height of the D-region Ionosphere

Figure 4 illustrates the mean electron fluxes and X-ray rates in high-latitude regions, both measured by CSES HEPP during nighttime and the h' value inferred from CSES EFD data, from 2019 to 2021. In Fig. 4, the first column corresponds to the Northern Hemisphere, and the second column to the Southern Hemisphere. Figures 4 (a-b) depict the distribution of the



nighttime mean integral electron flux for 100 keV–3 MeV ($\log_{10}$(1/cm²/s/sr)). At the CSES satellite's altitude of 507 km, there exists a distinct region with enhanced high-energy particle flux. This enhancement is closely associated with the L-value, approximately around $L = 5$, corresponding to the center of outer electron radiation (Reeves et al., 2016). Figures 4 (c-d) show the distribution of nighttime mean X-ray rate ($\log_{10}$(counts/s)). The distribution of the X-ray rate closely follows that of precipitation electrons, with higher X-ray rates measured exactly in regions of enhanced electron fluxes. The X-rays are generated through bremsstrahlung radiation of precipitation electrons with air molecules, which occurs deep in the atmosphere. The electrons that produce these bremsstrahlung X-rays are indeed precipitation ones. Figures 4 (e-f) show the distribution of the D-region ionosphere reflection height. In regions where electron and X-rays fluxes are enhanced, specifically in the region where the L-value is approximately 5, a noticeable decrease in reflection height is observed. This is very likely due to the fact of high-energy precipitating electrons ionizing atmospheric molecules in the D-region ionosphere.

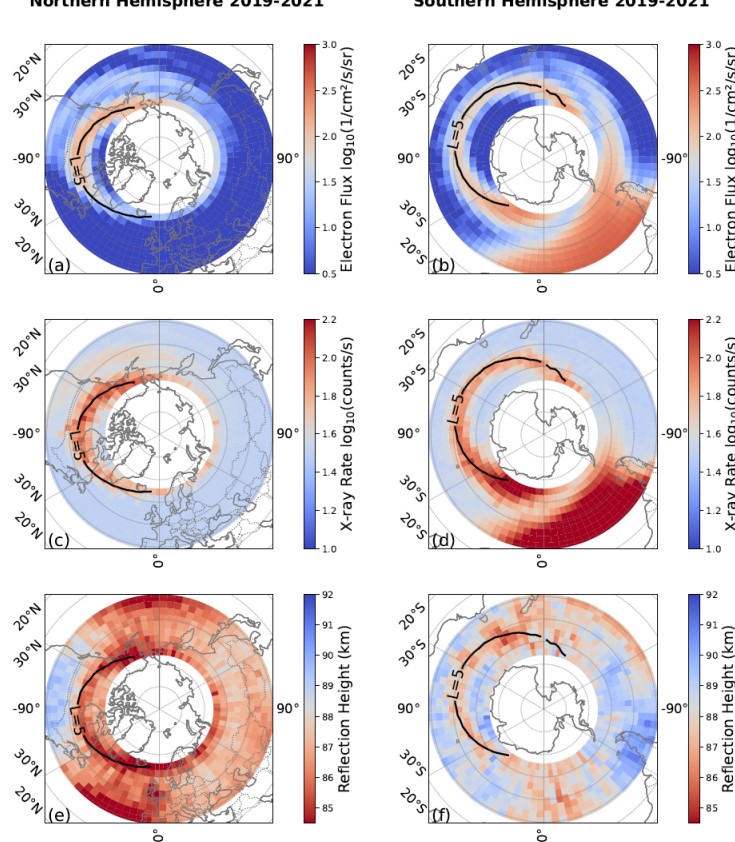



**Figure 4: (a) and (b) show the nighttime electron flux distributions for the Northern and Southern Hemispheres. (c) and (d) display the nighttime X-ray intensity distributions for the Northern and Southern Hemispheres. (e) and (f) show the nighttime reflection height of the D-region ionosphere for the Northern and Southern Hemispheres, respectively. The black line represents the region**
**with an L-value of 5. All data are the averages from 2019 to 2021.**

To quantitatively investigate the spatial correlations among electron flux, X-ray rate, and D-region reflection height, we performed a bivariate Moran's I analysis for the region where 3≤L≤10. This range was chosen because it is primarily influenced by high-energy particle precipitation from the outer radiation belt. Table 1 summarizes the results of this analysis.


**Table 1: Electron Flux, X-ray Rate and Reflection Height Moran's I relationship analysis (3<L<10)**

| Region | Data | Moran's I | P-Value |
|---|---|---|---|
| Northern Hemisphere | Flux & Reflection Height | -0.386 | 0.001 |
| | Flux & X-ray rate | 0.4468 | 0.001 |
| | Reflection Height & X-ray rate | -0.4176 | 0.001 |
| Southern Hemisphere (longitude: 0°–60°) | Flux & Reflection Height | -0.2455 | 0.013 |
| | Flux & X-ray rate | 0.6003 | 0.001 |
| | Reflection Height & X-ray rate | -0.2728 | 0.001 |
| Southern Hemisphere (longitude: 60°–180°) | Flux & Reflection Height | -0.4535 | 0.001 |
| | Flux & X-ray rate | 0.7869 | 0.001 |
| | Reflection Height & X-ray rate | -0.4308 | 0.001 |

Notably, in the Southern Hemisphere, we divided the region into 0°-60° (influenced by the South Atlantic Anomaly) and 60°-180° (not affected by the South Atlantic Anomaly), as will be explained in Sect. 3.2. After excluding the values affected by
the South Atlantic Anomaly, the overall trends and statistical significance ($p < 0.01$) show a high degree of consistency in both hemispheres. This highlights a strong global correlation among electron flux, X-ray rate, and D-region reflection height. Specifically, electron flux and X-ray rate are positively correlated, while both are negatively correlated with reflection height. Building on these correlation results and integrating three CSES measurements—electron flux, X-rays, and reflection height— we can construct a comprehensive physical scenario: High-energy electrons (100 keV–3 MeV) detected by HEPP-L at an
altitude of 507 km precipitate into the D region ionosphere (60–90 km). During precipitation, collisions with atmospheric molecules produce bremsstrahlung X-rays, which are backscattered and subsequently captured by the HEPP-X instrument, also at 507 km. Meanwhile, the precipitating electrons strengthen ionization in the D region, thereby lowering its reflection height. Moran's I results corroborate this mechanism, particularly around L=5, where electron precipitation is most intense, leading to notably increased ionization and a marked decrease in reflection height.






### 3.2 The Impact of Electron Flux on the Reflection Height of the D-region ionosphere

Figures 5 (a-h) show the 3-year average data from 2019 to 2021. Figures 5 (a-b) show the electron energy flux calculated within the loss cone. Figures 5 (c-d) represent the reflection height calculated using the cutoff frequency, with the gray solid line indicating the trend of the reflection height. Figures 5 (e-f) represent the reflection height modeled by the PyGPI5 model

fitted using the WS formula (Method 1 in Section 2.4), with the gray solid line indicating the trend of the reflection height and the gray dashed line representing the trend of the quiet period (without EEP) reflection height. Figures 5 (g-h) represent the reflection height calculated using Method 2 in Section 2.4, based on the simulated electron density and cutoff frequency. The red dots represent data from the entire longitude range of the Northern Hemisphere. The yellow and blue dots represent data from the Southern Hemisphere for the longitude ranges of 0-60° and 60-180°, respectively.

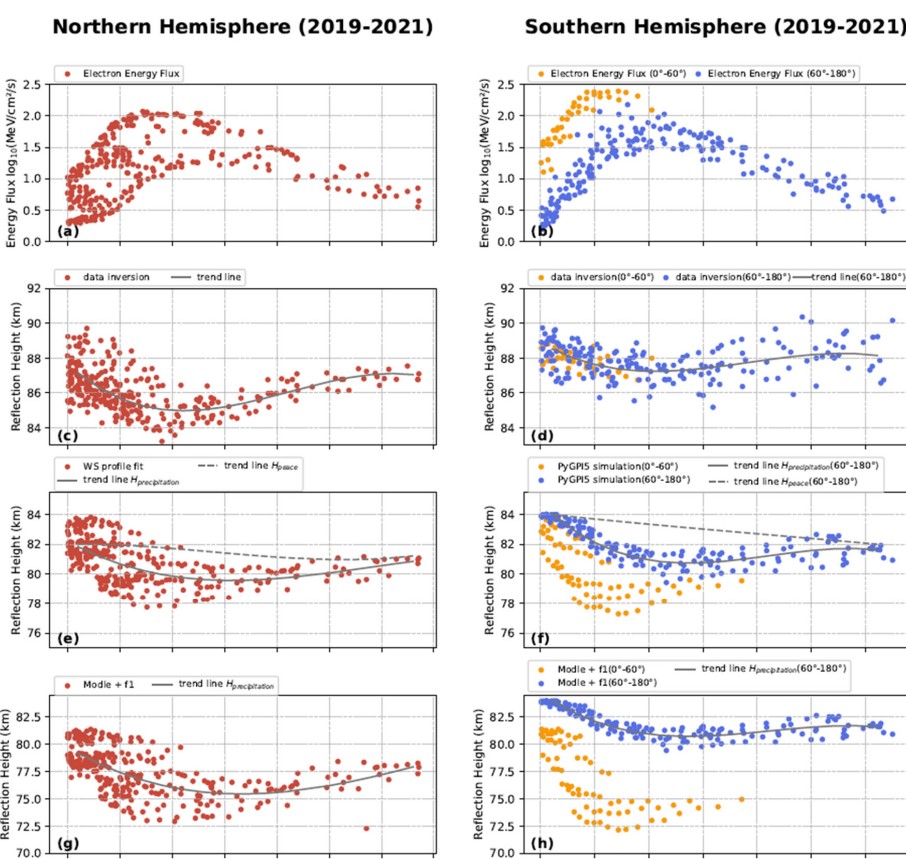




**Figure 5: (a-h) shows the 3-year average data from 2019 to 2021. (a-b) show the electron energy flux calculated within the loss cone. (c-d) represent the reflection height calculated using the cutoff frequency, with the gray solid line indicating the trend of the reflection height. (e-f) represent the reflection height modeled by the PyGPI5 model fitted using the WS formula (Method 1), with the gray solid line indicating the trend of the reflection height and the gray dashed line representing the trend of the quiet period (without**

**EEP) reflection height. (g-h) represent the reflection height calculated using Method 2, based on the simulated electron density and cutoff frequency. The red dots represent data from the entire longitude range of the Northern Hemisphere. The yellow and blue dots represent data from the Southern Hemisphere for the longitude ranges 0-60° and 60-180°, respectively.**

In Figs. 5 (a-b), it can be seen that the energy flux is mainly concentrated between L-values of 4 and 6, with the central region

around L=5, consistent with the electron distribution of the outer electron radiation belt (Baker et al., 2017). In the region where L-values are between 3 and 5, the energy flux increases; as observed in Figs. 5 (c-d), the reflection height in this area shows a significant decrease. Conversely, in the region where L-values are between 5 and 10, the energy flux decreases, leading to a rise in the reflection height. We have simulated the reflection height in the absence of precipitating electron flux as a reference, indicated by the gray dashed lines in Figs. 5 (e-f). In the Northern Hemisphere, across the full longitude range with

L-values from 4 to 7, the reflection height trends obtained from WS profile fitting (Method 1) and Ratcliffe's relation simulation (Method 2) closely match the results calculated from the cutoff frequency. The most significant decrease in reflection height occurs around L=5, which is consistent with previous studies (Chen et al., 2023). The PyGPI5 simulation found a reduction in reflection height of 2.5 km, which is almost identical to the $h'$ value derived from CSES data.

In the Southern Hemisphere, the complexity of the outer electron radiation belt is influenced by certain regions, particularly

the South Atlantic Anomaly (SAA). Therefore, we divided the Southern Hemisphere into two regions based on longitude: 0°-60° (influenced by the SAA) and 60°-180° (not influenced by the SAA). In the 0°-60° longitude range, the electron flux near the SAA is higher than in other regions. According to model simulations (yellow dots in Figs. 5 (f)) and (h)), the decrease in reflection height in this region is expected to be more significant. However, actual measurements show that the reflection height in this region is similar to that of the 60°-180° longitude range (blue dots in Fig. 5 (d)). This discrepancy was also noted

by Toledo-Redondo et al. (2012), who found that reflection-height contours do not fully align with the SAA's geographic boundaries. One possible explanation is that unique geomagnetic conditions in the SAA alter how precipitating electrons deposit energy, leading to complex and sometimes counterintuitive ionospheric responses. Further work is needed to disentangle these processes and quantify the net effect on the D-region ionosphere.

In the longitude range of 60° to 180°, the simulation results closely match the trend calculated from the cutoff frequency. The most significant decrease in reflection height occurs around L = 5, dropping by about 1.5 km (Fig. 5 (d)). The decrease in reflection height is more significant in the Northern Hemisphere compared to the Southern Hemisphere (60°-180° longitude). In the satellite observation region of the Northern Hemisphere, according to Fig. 6 of Anderson et al., 2018, within the radiation belt region covered by the CSES satellite's observation range [-65° to 65°], the electron counts are higher in the Northern

Hemisphere and lower in the Southern Hemisphere (Anderson et al., 2018).This higher flux leads to more ionization and thus



a reduction of the reflection height. As mentioned earlier, collisions of precipitated electrons with the neutral molecule can generate bremsstrahlung X-rays, so balloon or satellite-based X-ray measurements are used to estimate the fluxes of precipitating energetic electrons (Xu and Marshall, 2019). Therefore, we have also counted the mean X-ray rates in the above region in the Northern and Southern hemispheres using the X-ray detector on board the CSES satellite. The X-ray

measurements show that more ionization occurs in the Northern Hemisphere (mean X-ray rate is 78.8 counts/s), compared to the Southern Hemisphere (60°-180° longitude, mean X-ray rate is 71.2 counts/s). It should be noted that the observed and simulated reflection heights follow the same trend with L-values, but there is a height difference between the simulated and measured results in both hemispheres. The possible reason for the discrepancy is that the h' in the WS formula (Method 1) serves as only a rough approximation of the reflection height, rather than the actual reflection height. In the case of the Ratcliffe

relation simulation (Method 2), the difference in the calculated reflection height may stem from the oversimplified collision frequency formula, which does not account for variations in air density across different latitudes and longitudes. These variations can fluctuate significantly, sometimes by several hundred percent. Despite these limitations, both methods effectively capture the changes trend in electron density within the D-region of the ionosphere.

**3.3 Seasonal Variation of Reflection Height**

We divided the year into two sets of months to investigate seasonal variations in the reflection height. Specifically, we define the warm season for the Southern Hemisphere as November to March and for the Northern Hemisphere as May to September. Conversely, the cold season for the Northern Hemisphere is November to March, and for the Southern Hemisphere is May to September. Figure 6 displays the derived reflection heights under these seasonal conditions for both hemispheres from 2019

to 2021. In Figs. 6 (a–d), the black line marks the L-value of 5, which generally corresponds to the center of the outer radiation belt where high-energy electron precipitation is strongest. Figure 6 (a) shows the Southern Hemisphere's warm season, and Figure 6 (b) shows the Northern Hemisphere's warm season. Figures 6 (c-d) represent the Northern Hemisphere's cold season and the Southern Hemisphere's cold season, respectively.

From these plots, we see that during the cold season (Figs. 6 (c–d)), the reflection height decreases significantly around $L \approx 5$, where high-energy particle precipitation is most intense. In regions where $L > 5$ (where precipitation effects are weaker), the reflection height tends to increase. By contrast, in the warm season (Figs. 6 (a–b)), there is almost no noticeable change in the reflection height within the same L-value range. According to the magnetic mirror hypothesis, particles bounce between two magnetic mirror points (i.e., the Earth's poles), and thus the electron fluxes at conjugate points in the Northern and Southern

Hemispheres should be approximately the same. We examined the electron fluxes at an altitude of 507 km and found that the average electron flux trend for L-values between 4 and 7 (where the electron flux is higher) in both hemispheres is similar. Therefore, the observed seasonal changes in reflection height are unlikely to stem from differences in electron precipitation flux. Instead, they are more plausibly tied to variations in the background ionospheric electron density, which can differ substantially between summer and winter.





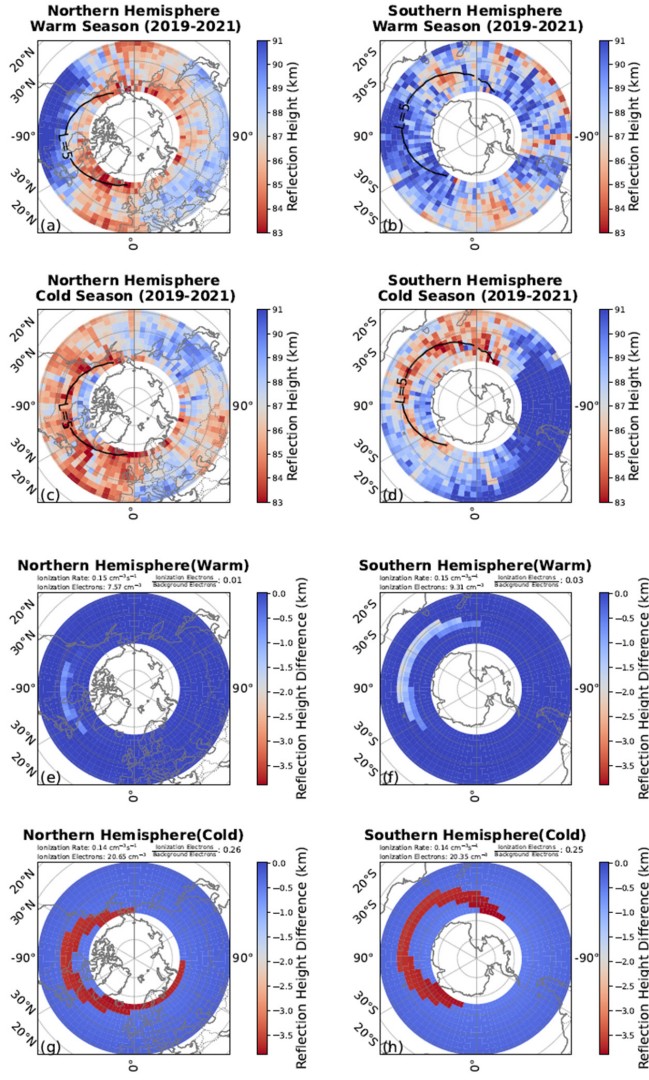


**Figure 6: (a-d) shows the derived reflection heights during the warm and cold seasons in both hemispheres from 2019 to 2021. (a) presents the Southern Hemisphere warm season, and (b) presents the Northern Hemisphere warm season. (c) represents the Northern Hemisphere cold season, and (d) represents the Southern Hemisphere cold season. The black line marks the L-value of 5. (e-h) shows the differences between the reflection height caused by particle precipitation and the quiet period reflection height during**

**warm and cold seasons, as simulated for both hemispheres. (e) and (f) show the reflection height differences for the Northern**



**Hemisphere in summer and the Southern Hemisphere in summer, respectively. (g) and (h) show the reflection height differences for the Northern Hemisphere in winter and the Southern Hemisphere in winter, respectively.**

To examine these seasonal differences more closely, Figures 6 (e-h) illustrate the differences between the reflection height caused by particle precipitation and that during quiet periods. These data were obtained using the PyGPI5 model WS profile (Method 1) simulation, as WS profile fitting can effectively model the reflection height under quiet conditions. To eliminate the influence of electron flux differences between the Northern and Southern Hemispheres, we selected the same energy flux as the precipitation input. The precipitation region was defined within the L-value range of 4 to 7, utilizing the energy spectrum, average energy flux, and a 1-hour integration time derived from CSES satellite observations between 2019 and 2021 in the L=4-7 range as standardized inputs. Figures 6 (e-h) use the year of 2020 as an example to show the reflection height variations for the Northern and Southern Hemispheres during summer and winter. Figure 6 (e) shows the difference for the Northern Hemisphere (summer) at 00:00 local time on July 1, 2020. Figure 6 (f) shows the difference for the Southern Hemisphere (summer) at 00:00 local time on January 1, 2020. Figure 6 (g) shows the difference for the Northern Hemisphere (winter) at 00:00 local time on January 1, 2020. Figure 6 (h) shows the difference for the Southern Hemisphere (winter) at 00:00 local time on July 1, 2020. Similar simulations were performed for the other two years, and the results were consistent with the trends observed in 2020.

It is evident from Figs. 6 (e–f) that in the warm seasons, the decrease in reflection height due to precipitation is smaller. In contrast, during the cold seasons (Figs. 6 (g–h)), the reflection height decrease is more substantial. We speculate this phenomenon is caused by the background electron density in summer and winter. To further clarify this mechanism, we focus on a location at 52.5° N, –90° E (L ≈ 5) and compare nighttime electron density variations on January 1 (winter) and July 1 (summer). In Fig. 7, we show the average ionization rate at 60–90 km (the collision region for incoming electrons) and the changes in electron density caused by ionization. Figure 7(a) reveal that while the ionized electron density in summer is slightly higher than in winter (because neutral densities are typically larger in summer, leading to more collisions), this increment is still not sufficient to cause a large change in reflection height. Figures 7 (b-c) illustrate how reflection height changes with the same ionization rate across different seasons. Notably, these changes are more pronounced in winter (Fig. 7b), where the background electron density is significantly lower compared to summer. When the same ionization rate is applied, a lower background electron density, as seen in winter, is more sensitive to changes caused by precipitation, resulting in greater perturbations. Once the electron density exceeds a certain threshold, further increases from precipitation have a diminished impact, which explains why summer experiences only minor changes in reflection height. Ultimately, the relatively low electron density in winter leads to stronger effects from precipitation and a more significant decrease in reflection height.



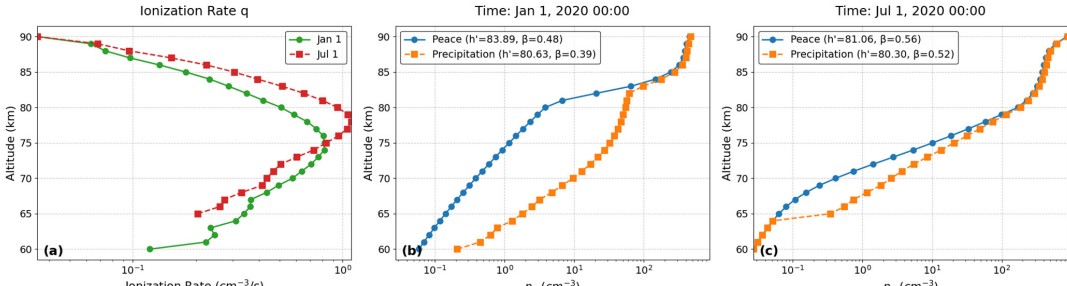

**Figure 7: Illustration of seasonal variations in electron density and reflection height at latitude 52.5° N, longitude –90° E (L ≈ 5). (a) Ionization rate at 00:00 LT on January 1 and July 1, 2020. (b, c) Electron density profiles during quiet conditions and after same**

**stable particle precipitation on January 1 and July 1, 2020, respectively.**

## 4. Summary

In this paper, using multi-payload collaborative observations from a single satellite, the CSES Satellite, we utilized observed electron flux, X-rays, and reflection height of ionosphere to reproduce the physical process in which radiation belt energetic electron precipitation (EEP) produce X-rays through bremsstrahlung radiation and alter the electron density in the D-region

ionosphere. An adapted version of the Toledo-Redondo et al. (2012) method was used to calculate the cutoff frequencies to estimate the reflection height of the D-region ionosphere in the high-latitude regions affected by EEP. Our results show that the influence of EEP on the two hemispheres is asymmetric: the reflection height in the Northern Hemisphere is in general lowered by 2.5 km, while the reflection height in the Southern Hemisphere is lowered by 1.5 km throughout the year. The decrease in reflection height exhibits seasonal variability in both hemispheres—being stronger in winter and weaker in summer.

This seasonal difference is likely related to the variation of the background ionospheric electron density.

**Code and data availability**

The PyGPI5 Model presented in this study can be found in online repositories. The names of the repository/repositories and accession number(s) can be found below: https://github.com/srkaeppler/pyGPI5. The CSES satellite data can be downloaded

on the website (https://www.leos.ac.cn/#/home) after registered (https://www.leos.ac.cn/#/register).



**Author contributions**

SZ and LL conceived the idea of the article. XD, SZ, and LL conducted the simulation, data analysis, interpretation, and manuscript preparation. XS provided the resources and commented on the paper. WX reviewed and contributed to the final paper draft and advised on interpreting the results. RL reviewed and contributed to the final paper draft and supervised the

calculation of EEP. XS discussed the result of the calculation of EEP. SH, YC, JL, and HL prepared data sets of CSES satellites. XD wrote the original paper, with edits from all other authors.

**Competing interests**

The contact author has declared that none of the authors has any competing interests.

**Financial support**

This work made use of the data from the CSES mission, a project funded by the China National Space Administration (CNSA) and the China Earthquake Administration (CEA). Thanks to the CSES satellite team for the data. This Project is supported by the Specialized Research Fund of the National Space Science Center, Chinese Academy of Sciences (Grant E4PD3010), National Natural Science Foundation of China (Grant 42274205), CAS Talents Program, Talent startup research Grants from

the National Space Science Center, Chinese Academy of Sciences (Grant 2023000034, E3RC2TQ4, E3RC2TQ5), the Specialized Research Fund for State Key Laboratories, National Space Science Center, Chinese Academy of Sciences (Grant E4262AA1), the China Research Institute of Radiowave Propagation (Research on low ionosphere satellite detection).

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
