# Peer review of "Quantitative Analysis of Nighttime Effects of Radiation Belt Energetic Electron Precipitation on the D-Region Ionosphere during Lower Solar Activity Periods"

_EGUsphere, 2025_

## Referee Comment (RC1)

REVIEW

**Quantitative Analysis of Nighttime Effects of Radiation Belt Energetic Electron Precipitation on the D-Region Ionosphere during Lower Solar Activity Period**

This study investigates the quantitative influence of EPP on the reflection height of ELF waves in the high latitudes of both hemispheres at low solar activity, considering the season. For the analysis, the wave and particle data measured by the Electric Field Detector (EFD) and High-Energy Particle Detector (HEPP) onboard the CSES-01 satellite during nighttime conditions between 2019 and 2021 were used. The reflection height is computed by applying two methods. The first approach derives the reflection height from the Wait and Spies formula by fitting the parameters "reflection height" and "sharpness" to a given electron density profile. In the second approach, the reflection height is computed using the cutoff frequency deduced from the spectral data obtained from the ELF band of the CSES satellite's EFD. Furthermore, the relationship between electron flux, X-ray rate, and D-region reflection height is investigated. The analysis revealed seasonal differences in the reflection heights, with lower reflection heights in winter than in summer. Moreover, an asymmetric influence of the EPP on the reflection height between the two hemispheres was found.

This work significantly contributes to understanding the influence of EPP on D-region ionization. The considerations of the seasons and the high latitudes in both hemispheres deliver new insights and raise questions for further investigations.

While the idea and approach of this study are very promising, the text is often imprecise and incomprehensible. The equations do not have a consistent notation. The structure of some figures is confusing, and the text information in the figures is not always readable due to a too-small font. Sometimes, there are just text segments (instead of complete sentences) or text duplications, giving an impression of inaccuracy. Before recommending it for publication, major revisions are suggested to be addressed. Please note the following points are listed in order of occurrence and not in order of minor and major revisions. Some points may appear minor and petty. However, the number of inaccuracies and inconsistencies sums up to a major category.

1. Line 70: *"However, the study on the global climatological characteristics of EEP effects is insufficient."* Does this statement refer to the study of Renkwitz et al., 2023? If yes, it would be good to be more specific. Why is the study insufficient? Generally, the sentence could also be removed, as it contains no information.
2. Line 115: Fang et al. (2008, 2010) are referred to twice in one sentence. One time is enough.
3. Line 120: reference style - *"Fang2010"*
4. Line 121: *"The Chemistry class contains the core of the GPI5 model (Glukhov et al., 1992)."* The GPI5 model needs to be introduced. In line 113, PyGPI is introduced, which is the Python implementation of the GPI5 D-region and ionization chemistry model. Using the terms PyGPI5 and GPI5 without a clear explanation makes it sound like two models.
5. Line 125: *"After inputting the ionization rate and precipitation duration, the model can produce the electron density profiles for a given location."* Where did you get the ionization rate?
6. Line 130: sentence duplication: *"After inputting the ionization rate and precipitation duration, the model can produce the electron density profiles for a given location"*
7. Line 137: Please explain in some words the Liouville's theorem and how the equation (1) was derived from it.
8. Line 140: *"By calculating the loss cone angle at satellite altitudes (507 km, $\propto_{LC}$)"* Please insert the equation for loss cone angle.
9. Line 147: *"To determine the cutoff frequency, we have improved the method of Toledo-Redondo et al. (2012), making it applicable to high-latitude regions."* This sentence is

confusing in this location. It can be removed since the topic is addressed later in the text (lines 163-165).

10. Line 152: f1=c/h' reference? In Saini (2010) the first order cut-off frequency looks different: fc = c/(2*h') ( **https://doi.org/10.1029/2009JA014795**  What causes the difference?

11. Line 155: *"An increase in electron density lowers the reflection height."* Inserting a reference would strengthen the statement.

12. Line 160: *"…and their downward propagation can influence the determination of the cutoff frequency."* Inserting a reference would strengthen the statement.

13. Line 149: Please explain in short words QTM1 and QTME.

14. Line 170: What window/kernel size and iteration step did you use? Why did you choose that specific window/kernel size? Please be more specific; the description is quite general.

15. Line 172-176: Why did you choose a different approach in the second step? Why didn't you use fftconvolve in the first step since it is more computationally efficient (as it is written in lines 174-175)?

16. Line 176: "After these two steps, the trend line of the ELF wave is extracted." The sentence is confusing. If I have understood correctly, the smoothed line is the trend line. However, it sounds like there is an extra extraction step. The term trend line is also confusing, as it is just the smoothed line.

17. Figure 1, caption: *"…and the red dots indicate the cutoff frequency…"* There is only one red dot.

18. Line 191: *"…(Shown in Fig. 2)"* No capitalisation at this point.

19. Figure 2: Confusing order of panels. 1500 keV is larger than 1 MeV; why is it in the middle? Switching between 1 MeV and 1500 keV is not consistent. Either 1000 keV and 1500 keV or 1 MeV and 1.5 MeV.

20. Please add some short words about what information we gain from Figure 2. Why did you choose 1 MeV and 1.5 MeV, as they are so close and there is no significant difference?  Why is it essential for the study?

21. Line 212-214: There is a dot and a space in the brackets by mistake. It would be nice if equation (3) showed the formula described. However, it shows the formula that has already been rearranged according to the plasma frequency (it is not wrong, but inaccurate).

22. Line 216: Where is the h' in the equation? Is z=h'? Please add a reference for equation 4.

23. Line 221: Please insert reference for equation (5).

24. In equation (6) it says z and not h'. Notation is not consistent.

25. Line 225-226: *"Here, Ne is ==the corresponding== electron density (in cm-3), which can be obtained from the PyGPI5 simulations, ƒ1 is the cutoff frequency calculated from modelled electron density profile==. derived from observations, h' is the reflection height (in km)."*==
    Corresponding to an altitude (->Ne(z))?
    The second yellow section is no sentence.

26. Figure 3: Same problem as in Figure 2. Confusing order of panels and switching between keV and MeV.

27. Line 265-267: *"The X-rays are generated through bremsstrahlung radiation of precipitation electrons with air molecules, which occurs deep in the atmosphere. The electrons that produce these bremsstrahlung X-rays are indeed precipitation ones."* Please rephrase the sentences and add references.

28. Lines 263-269: The discussion should also mention the low reflection height in 0°-(45°) longitude in Fig. 4e, whereas there is no notable X-ray rate in this area (Fig. 4c). Moreover, the discussion should include the striking high X-ray rate from 0°-70° longitude in Fig. 4d, while the reflection height is only slightly decreased from 45-90° longitude (Fig. 4f).

29. Line 279: Please describe Moran's I shortly. Why did you choose that approach, and what information is provided (e.g., Positive/negative correlation-> cumulation, dispersion, information from the value of Moran score)?

30. Line 294: A comparison with other studies would strengthen the conclusion.

31. Figure 5: The font in the legend is too small. The text is not readable.

32. Line 345-346: *"The X-ray measurements show that more ionization occurs in the Northern Hemisphere (mean X-ray rate is 78.8 counts/s), compared to the Southern Hemisphere (60°-180° longitude, mean X-ray rate is 71.2 counts/s)."* Where did you get the X-ray rates?

33. Line 348-349: *"The possible reason for the discrepancy is that the h' in the WS formula (Method 1) serves as only a rough approximation of the reflection height, rather than the actual reflection height."* Could one possible reason be that the Wait and Spies formula refers to VLF/LF waves? You compare with reflection heights of ELF waves, which, assumably, are lower by nature. Perhaps comparing the qualitative behavior as a function of the EPP at different energies is more meaningful than looking at the quantitative differences.

34. Line 352*: "These variations can fluctuate significantly, sometimes by several hundred percent."* Please insert a reference. Furthermore, it should be discussed, that both methods use the electron density profile from PyGDI5 what bases on IRI. Does IRI have any limitations?

35. Figure 6 e-h: The font of the information test below the title is too small. Text not readable.

36. Figure 7: The caption says "quiet conditions," but the legend denotes "Peace." Please be more consistent with the notation. I prefer quiet conditions.

---

## Author Comment (AC1)

Response to Reviewer Comment

We greatly appreciate the reviewer's valuable comments and detailed feedback on our manuscript. The issues and suggestions raised have been instrumental in improving the overall quality and clarity of our work. We have carefully reviewed all comments and made comprehensive revisions, including unifying the notation used in equations, enhancing the readability of figures, and refining the text for greater clarity and accuracy. We are especially grateful for the reviewer's recognition of the scientific significance of our study on the effects of Energetic Electron Precipitation (EEP) on D-region ionization. We believe the revised manuscript has been significantly improved in both quality and readability.

1. Line 70: "However, the study on the global climatological characteristics of EEP effects is insufficient." Does this statement refer to the study of Renkwitz et al., 2023? If yes, it would be good to be more specific. Why is the study insufficient? Generally, the sentence could also be removed, as it contains no information.

We have removed this statement as suggested by the reviewer, as it does not add substantial information to the manuscript and may cause confusion regarding the work of Renkwitz et al., 2023.

2. Line 115: Fang et al. (2008, 2010) are referred to twice in one sentence. One time is enough.

We have rewritten this sentence to eliminate the redundancy in Line 114 of the revised manuscript: "The Ionization class utilizes the models developed by Fang et al. (2008, 2010) to generate altitude profiles of ionization for ions and electrons."

3. Line 120: Reference style - *"Fang2010"*

We have corrected the inconsistent reference style to follow the journal's format in Lines 120 of the revised manuscript: "by inputting monoenergetic energy (keV), electron energy flux (erg/cm²/s), geographic coordinates, and the desired altitude range into the *"Fang2010"* model."

4. Line 121: "The Chemistry class contains the core of the GPI5 model (Glukhov et al., 1992)." The GPI5 model needs to be introduced. In line 113, PyGPI is introduced, which is the Python implementation of the GPI5 D-region and ionization chemistry model. Using the terms PyGPI5 and GPI5 without a clear explanation makes it sound like two models.

We have clarified the relationship between PyGPI5 and GPI5 in our manuscript. PyGPI5 consists of two main components: the Ionization class (*"Fang2010"* model) and the Chemistry class (GPI5 model). The workflow first uses the Fang2010 model in the Ionization class to calculate the ionization rate, which is then input into the GPI5 model to obtain the electron density profile. We have already made this clarification in lines 125-127 of our manuscript: "After inputting the ionization rate obtained from the *"Fang2010"* model and precipitation duration, the GPI5 model can produce the electron density profiles for a given location."

5. Line 125: "After inputting the ionization rate and precipitation duration, the model can produce the electron density profiles for a given location." Where did you get the ionization rate?

We have clarified the source of the ionization rate: "After inputting the ionization rate obtained from the *"Fang2010"* model and precipitation duration, the GPI5 model can produce the electron density profiles for a given location." We have revised the text in Lines 125-127.

6. Line 130: sentence duplication: "After inputting the ionization rate and precipitation duration, the model can produce the electron density profiles for a given location"
We have removed the duplicated sentence as pointed out by the reviewer.

7. Line 137: Please explain in some words the Liouville's theorem and how the equation (1) was derived from it.
Liouville's Theorem in Statistical Mechanics: In conservative systems, the phase space density remains constant along particle dynamical trajectories.

In space physics applications, this theorem leads to the important relationship between pitch angle ($\theta$) distribution and magnetic field strength (B): $\sin^2\theta/B$ = constant, which forms the theoretical foundation for studying particle loss cone and particle trapping phenomena in space physics.

8. Line 140: "By calculating the loss cone angle at satellite altitudes (507 km, $\propto LC$)" Please insert the equation for loss cone angle.
Based on the pitch angle ($\alpha_{507km}$) measured by the satellite and the magnetic field intensities ($B_{507km}$, $B_{100km}$) obtained from the IGRF model, we calculate α_100km by applying Liouville's theorem. This calculation is presented in Equation 1, and we have accordingly modified previously ambiguous explanations to clarify this process. The revised text now reads: "By measuring pitch angle at the satellite altitude (507 km) and calculating the loss cone angle at atmosphere altitudes (100 km, $\alpha_{LC}$), we can determine the electron fluxes that can precipitate into the atmosphere." We have revised the text in Lines 137-138.

9. Line 147: "To determine the cutoff frequency, we have improved the method of Toledo-Redondo et al. (2012), making it applicable to high-latitude regions." This sentence is confusing in this location. It can be removed since the topic is addressed later in the text (lines 163-165).
We have removed the confusing sentence as suggested, as the topic is addressed later in the text.

10. Line 152: f1=c/h' reference? In Saini (2010) the first order cut-off frequency looks different: fc = c/(2*h') ( https://doi.org/10.1029/2009JA014795 What causes the difference?
I apologize for the error. You're absolutely right about the first-order cutoff frequency formula. The correct formula is indeed fc = c/(2*h'), as properly referenced in Saini (2010). Thank you for pointing out this discrepancy. The formula should be corrected to fc = c/(2*h') to be consistent with the established literature in Saini (2010) and other sources. This was my typographical error, but I did actually use the correct formula fc = c/(2*h') when calculating the cutoff frequency in the computations. We have revised the text in Line 148.

11. Line 155: "An increase in electron density lowers the reflection height." Inserting a reference would strengthen the statement.

We have added a reference: "An increase in electron density lowers the reflection height (Cheng et al., 2023; Gasdia and Marshall, 2023)." We have revised the text in Lines 150-151.

12. Line 160: "…and their downward propagation can influence the determination of the cutoff frequency." Inserting a reference would strengthen the statement.

We have added a reference: "In the magnetosphere, auroral hiss, chorus waves, and lower hybrid electrostatic noise also exist, and their downward propagation can influence the determination of the cutoff frequency (Martinez-Calderon et al., 2015; Yu et al., 2023). " We have revised the text in Lines 155-156.

13. Line 149: Please explain in short words QTM1 and QTME.

QTM1 and QTEM are two characteristic electromagnetic wave propagation modes in the Earth-ionosphere waveguide, with their key differences primarily manifested in frequency range applicability and field structure. The QTM1 mode primarily operates in the extremely low frequency (ELF) band of 300Hz to 1.8kHz, featuring horizontally distributed magnetic fields and significantly vertical electric field components. This unique field configuration enables exceptionally low propagation loss, allowing ultra-long-distance transmission over thousands of kilometers with the capability to penetrate seawater, making it widely applicable in military domains such as submarine communications. In contrast, the QTEM mode dominates the high-frequency band above 1.8kHz, with both electric and magnetic fields exhibiting horizontal distribution, and its propagation characteristics more closely resemble free-space waves. Although it offers relatively shorter propagation distances and greater attenuation, it is better suited for medium-to-short-range communication needs such as shortwave broadcasting and aeronautical communications. These two modes exhibit a distinct transition zone around 1.6-1.8kHz, where this critical frequency fluctuates with solar activity and geomagnetic field variations. We insert references in Line 146 to strengthen the statement (Toledo-Redondo et al., 2012; Ramo et al., 1994).

14. Line 170: What window/kernel size and iteration step did you use? Why did you choose that specific window/kernel size? Please be more specific; the description is quite general.

We used a window/kernel size of 45 for our analysis. This parameter was determined through repeated experimental testing of the data. The window size of 45 was chosen because it effectively locates the minimum points in the waveform while avoiding excessive smoothing of the wave trend line. Smaller windows might result in noise interference, while larger windows could cause important features to be over-smoothed and lost.

15. Line 172-176: Why did you choose a different approach in the second step? Why didn't you use fftconvolve in the first step since it is more computationally efficient (as it is written in lines 174-175)?

We sincerely appreciate the reviewer's insightful question. Regarding the rationale behind employing a two-step smoothing approach (using convolve in the first step and fftconvolve in

the second), the core objective of this processing workflow is to accurately identify the local minima of ELF wave energy (i.e., the cutoff frequency). Our detailed explanation is as follows:

The initial step utilizes convolve for moving average smoothing, which effectively removes high-frequency noise while preliminarily extracting the primary signal trend. However, the results from this first-step smoothing alone are insufficient to precisely determine the desired trendline (particularly for identifying energy minima), as residual noise or local fluctuations may still interfere with minimum value detection. Therefore, we implement a secondary processing step employing Fast Fourier Transform-based convolution (fftconvolve) on the initially smoothed data to more efficiently suppress residual noise and ensure both the smoothness and reliability of the trendline.

The ELF wave energy minima (cutoff frequencies) extracted through this method demonstrate excellent agreement with the global statistical results obtained by Toledo-Redondo et al. using DEMETER satellite data (as shown in Figure 1). This consistency validates the effectiveness of our approach, confirming that the two-step smoothing strategy successfully preserves the physical characteristics of the signal while meeting practical application requirements.

We will further emphasize the importance of this method in accurately determining energy minima in the manuscript and make revisions in Lines 165-169.

16. Line 176: Trend line extraction clarification
Thank you for your correction. This sentence is indeed unclear. Your understanding is correct - the smoothed line is the trend line, and there is no additional extraction step. The sentence can be revised to: "After these two smoothing steps, we obtain the smoothed curve of the ELF wave." which is more direct and clear. The term "trend line" can indeed cause confusion, as it is simply the smoothed curve obtained through the two-step smoothing process. We have revised the text in Lines 169-170.

17. Figure 1, caption: "…and the red dots indicate the cutoff frequency…" There is only one red dot.
Thank you for your correction. We have revised the caption.

18. Line 191: "…(Shown in Fig. 2)" No capitalisation at this point.
Thank you for the feedback. The capitalization issue has been corrected in Line 184.

19. Figure 2: Confusing order of panels. 1500 keV is larger than 1 MeV; why is it in the middle? Switching between 1 MeV and 1500 keV is not consistent. Either 1000 keV and 1500 keV or 1 MeV and 1.5 MeV.
Thank you for your feedback. The issue with the panel order in Figure 2 has been addressed, and the sequence has been corrected for consistency. The 1500 keV has been revised as 1.5 MeV and is now placed in the correct position relative to 1 MeV, ensuring clarity. We appreciate your suggestion and have made the necessary adjustments.

20. Please add some short words about what information we gain from Figure 2. Why did you choose 1 MeV and 1.5 MeV, as they are so close and there is no significant difference? Why is it essential for the study?

Figure 2 is designed to verify the temporal variations of electrons at different energy levels and to demonstrate that the effects of particle precipitation reach a steady state after 3600s. This verification is crucial for our study because electrons at different energy levels affect ionization rates at different altitudes. The observed minimal difference between 1 MeV and 1.5 MeV electrons can be attributed to their relatively similar altitude ranges of influence and ionization effects. Therefore, it can be inferred that the effect of higher energy level electrons on the electron density in the D-region may be closer to that of 1 MeV electrons.

21. Line 212-214: There is a dot and a space in the brackets by mistake. It would be nice if equation (3) showed the formula described. However, it shows the formula that has already been rearranged according to the plasma frequency (it is not wrong, but inaccurate).

We have removed the extra dot and space, thank you for your suggestion.

22. Line 216: Where is the h' in the equation? Is z=h'? Please add a reference for equation 4.
23. Line 221: Please insert reference for equation (5).
24. In equation (6) it says z and not h'. Notation is not consistent.
25. Line 225-226: "Here, $N$ e is the corresponding electron density (in cm-3), which can be obtained from the PyGPI5 simulations, $f$ 1 is the cutoff frequency calculated from modelled electron density profile. derived from observations, $h$ ' is the reflection height (in km)."

We sincerely appreciate the reviewer's careful reading and constructive suggestions. We have revised the relevant text as follows to improve clarity and precision:

In the equation (4) and (6), z refers to the general altitude, while h' represents the reflection height at a specific frequency. We have corrected the issue of z and h' in the text and inserted references. The revised content for lines 213-214: "and z is the altitude in kilometres. when discussing specific reflection heights at a particular frequency, h' will be used." The revised content for lines 220-223: " In this formulation, $Ne$(z) represents the electron density (in cm$^{-3}$) at reflection height z, obtained from PyGPI5 simulations. The cutoff frequency $f$ 1 corresponds to the wave frequency that is reflected at a specific height z. Using equation 6, h' can be determined by finding the height z at which the calculated cutoff frequency matches the observed $f_1$ from CSES EFD measurements."

26. Figure 3: Same problem as in Figure 2. Confusing order of panels and switching between keV and MeV.

We sincerely appreciate the reviewer's valuable feedback regarding Figure 3. We have carefully addressed the concerns about panel organization and energy unit consistency through the modifications.

27. Line 265-267: "The X-rays are generated through bremsstrahlung radiation of precipitation electrons with air molecules, which occurs deep in the atmosphere. The electrons that produce these bremsstrahlung X-rays are indeed precipitation ones." Please rephrase the sentences and add references.

We have modified the sentences to read: " The X-rays are generated through bremsstrahlung radiation of precipitation electrons with air molecules, which occurs deep in the atmosphere (Xu et al., 2020)." We have revised the text in Lines 261-263.

28. Lines 263-269: The discussion should also mention the low reflection height in 0°-(45°) longitude in Fig. 4e, whereas there is no notable X-ray rate in this area (Fig. 4c). Moreover, the discussion should include the striking high X-ray rate from 0°-70° longitude in Fig. 4d, while the reflection height is only slightly decreased from 45-90° longitude (Fig. 4f).
We thank the reviewer for this insightful observation. We have incorporated a detailed discussion of the phenomenon mentioned in the 0°-45° longitude region where there is low reflection height (Fig. 4e) despite no notable X-ray rate (Fig. 4c).

we have further addressed the phenomenon in the 0°–45° longitude region in Lines 280-282: "In the Northern Hemisphere, within the region of 0°–45° longitude and 40°–50° latitude, the underlying reason is that this area lies within the Inter-Tropical Convergence Zone, where the ionosphere is highly variable and exhibits the highest occurrence of plasma bubbles (Kil and Heelis, 1998; Su et al., 2006; Toledo-Redondo et al., 2012)."

Regarding the strikingly high X-ray flux observed in the 0°–70° longitude region, we believe this phenomenon may be related to the South Atlantic Anomaly (SAA). In the observations from the CSES satellite at an altitude of 507 km, we detected a significant enhancement in electron flux within this region. As shown in Fig. 1, both high-energy electrons (2.9–3 MeV) and medium-energy electrons (0.5–0.7 MeV) exhibit strong fluxes within the 0°–70° longitude range. We hypothesize that this may be associated with the weakened geomagnetic field within the SAA region.

[Figure]

[Figure]

Figure 1. Average Electron Flux in 2019

Moreover, according to the study by Toledo et al. (2012), the SAA does not show significant changes in reflection height during high-energy electron precipitation events. Therefore, the mechanisms potentially influencing the 0°–70° longitude region near the SAA may be similar to those observed within the SAA itself. Given the complexity of the SAA, we divided the Southern Hemisphere into two regions for analysis: 0°–60° longitude (affected by the SAA) and 60°–180° longitude (unaffected by the SAA). As for why the reflection height in the 0°–60° region is only slightly decreased despite the strong electron and X-ray flux, we have included this phenomenon in our future research plans. Further simulations and data analysis will be conducted to explore the underlying mechanisms.

29. Line 279: Please describe Moran's I shortly. Why did you choose that approach, and what information is provided (e.g., Positive/negative correlation-> cumulation, dispersion, information from the value of Moran score)?

Moran's I is a spatial autocorrelation statistical method used to measure the degree of clustering or dispersion of data in space. We chose this method because it effectively quantifies the correlation between X rays, precipitated electrons and reflection height in the spatial distribution.

The values of Moran's I typically range from -1 to +1:

- Positive values (close to +1) indicate positive correlation.
- Negative values (close to -1) indicate negative correlation.
- Values close to zero indicate no apparent correlation.

We have revised Lines 274-275

30. Line 294: A comparison with other studies would strengthen the conclusion.

We appreciate this valuable suggestion and have added comparisons with several related studies to strengthen our conclusions regarding the relationship between electron flux, X-ray intensity, and changes in reflection height. The following paragraph has been added in Lines 294-300: " PyGPI5 model simulations show that energetic electrons precipitate to altitudes of 60-90 km (Fang et al., 2010). Xu and Marshall (2019) demonstrated that these precipitating electrons produce bremsstrahlung X-rays which can be backscattered and detected by satellites, while also increasing the electron density in the lower ionosphere. This increase in electron density leads to a decrease in the reflection height, which aligns with our observations. Chen et al.

(2023) quantitatively analyzed the global changes in electron density through a simulation. Our study combined the satellite observation of electron precipitation, X-ray, and ionospheric reflection height changes, and these three variables showed high correlation through spatial correlation analysis."

31. Figure 5: The font in the legend is too small. The text is not readable.
We sincerely appreciate the reviewer's careful observation regarding Figure 5. We have made the improvements to address the readability issues.

32. Line 345-346: "The X-ray measurements show that more ionization occurs in the Northern Hemisphere (mean X-ray rate is 78.8 counts/s), compared to the Southern Hemisphere ($60°$-$180°$ longitude, mean X-ray rate is 71.2 counts/s)." Where did you get the X-ray rates?
We appreciate the reviewer's question regarding the X-ray rates mentioned in Lines 350–351. "Therefore, we also calculated the average X-ray rates in the above-mentioned region of the Northern and Southern Hemispheres using the X-ray detector onboard the CSES satellite.."

33. Line 348-349: "The possible reason for the discrepancy is that the h' in the WS formula (Method 1) serves as only a rough approximation of the reflection height, rather than the actual reflection height." Could one possible reason be that the Wait and Spies formula refers to VLF/LF waves? You compare with reflection heights of ELF waves, which, assumably, are lower by nature. Perhaps comparing the qualitative behavior as a function of the EPP at different energies is more meaningful than lookin at the quantitative differences.
The WS (Wait and Spies) formula is an empirical model used to fit the electron density profile of the ionospheric D-region. The reflection height h' defined in the formula does not represent the actual reflection height of electromagnetic waves, but rather serves as a fitting parameter to derive the electron density distribution. Therefore, there is indeed a discrepancy between the defined h' and the actual wave reflection height. However, since the WS formula provides a good approximation of the D-region electron density profile, the estimated h' still shows qualitative consistency with the actual reflection height.

In addition, the reviewer made an excellent point that the WS formula was originally developed for VLF/LF wave propagation, whereas in our study we are discussing the reflection heights of ELF waves, which are generally lower by nature. This difference in wavebands could indeed contribute to the quantitative discrepancy. As suggested, comparing the qualitative behavior of reflection height as a function of energetic particle precipitation (EPP) at different energies may be more meaningful than focusing solely on quantitative differences. We sincerely appreciate the reviewer's valuable suggestion and plan to explore this aspect further in future research.

34. Line 352: "These variations can fluctuate significantly, sometimes by several hundred percent." Please insert a reference. Furthermore, it should be discussed, that both methods use the electron density profile from PyGDI5 what bases on IRI. Does IRI have any limitations?
We sincerely appreciate the valuable comments provided by the reviewer. We have inserted the references as requested. Additionally, we acknowledge the limitations of the IRI model when used as the basis for the electron density profile in PyGPI5. While IRI performs well in many

applications, it has known limitations in the lower ionosphere region. For example, the IRI model struggles to capture the fine spatial variations in electron density at low altitudes, particularly with respect to geographic location (Toledo et al., 2012). Moreover, during solar activity peaks, the IRI model may not fully reflect the dynamic response of the lower ionosphere to solar cycle variations (Zhao et al., 2024). These model limitations may be one of the reasons for the discrepancies observed in our simulations, and we have discussed this in the revised manuscript in Lines 358-365: "These variations can fluctuate significantly, sometimes by several hundred percent (Sheese et al., 2011; Emmert et al., 2021). Additionally, both methods rely on the electron density profile obtained from the PyGPI5 model, which is based on the IRI (International Reference Ionosphere) model. However, the IRI model has limitations, particularly in capturing the variations in electron density in the lower ionosphere. For example, IRI struggles to accurately reflect the spatial variations in electron density with respect to latitude and longitude, especially in the lower ionospheric layers (Toledo-Redondo et al., 2012). Furthermore, the IRI model's representation of electron density in the lower ionosphere may not fully capture the dynamic changes associated with the solar cycle, which could influence the results (Zhao et al., 2024)."

35. Figure 6 e-h: The font of the information test below the title is too small. Text not readable. Thank you for your comment. We have increased the font size of the text below the title to ensure it is now readable.

36. Figure 7: The caption says "quiet conditions," but the legend denotes "Peace." Please be more consistent with the notation. I prefer quiet conditions. We appreciate your feedback. We have updated the legend to read "quiet conditions" to maintain consistency with the caption.

References

Cheng, W., Xu, W., Gu, X., Wang, S., Wang, Q., Ni, B., Lu, Z., Xiao, B., and Meng, X.: A Comparative Study of VLF Transmitter Signal Measurements and Simulations during Two Solar Eclipse Events, Remote Sensing, 15, 3025, 2023.

Emmert, J. T., Drob, D. P., Picone, J. M., Siskind, D. E., Jones Jr., M., Mlynczak, M. G., Bernath, P. F., Chu, X., Doornbos, E., Funke, B., Goncharenko, L. P., Hervig, M. E., Schwartz, M. J., Sheese, P. E., Vargas, F., Williams, B. P., and Yuan, T.: NRLMSIS 2.0: A Whole-Atmosphere Empirical Model of Temperature and Neutral Species Densities, Earth and Space Science, 8, e2020EA001321, https://doi.org/10.1029/2020EA001321, 2021.

Fang, X., Randall, C. E., Lummerzheim, D., Wang, W., Lu, G., Solomon, S. C., and Frahm, R. A.: Parameterization of monoenergetic electron impact ionization, Geophysical Research Letters, 37, https://doi.org/10.1029/2010GL045406, 2010.

Gasdia, F. and Marshall, R. A.: A Method for Imaging Energetic Particle Precipitation With Subionospheric VLF Signals, Earth and Space Science, 10, e2022EA002460, https://doi.org/10.1029/2022EA002460, 2023.

Kil, H. and Heelis, R. A.: Global distribution of density irregularities in the equatorial ionosphere, Journal of Geophysical Research: Space Physics, 103, 407-417, https://doi.org/10.1029/97JA02698, 1998.

Martinez-Calderon, C., Shiokawa, K., Miyoshi, Y., Ozaki, M., Schofield, I., and Connors, M.: Polarization analysis of VLF/ELF waves observed at subauroral latitudes during the VLF-CHAIN campaign, Earth, Planets and Space, 67, 21, 10.1186/s40623-014-0178-7, 2015.

Ramo, S., Whinnery, J. R., and Van Duzer, T.: Fields and Waves in Communication Electronics, Wiley1994.

Sheese, P. E., McDade, I. C., Gattinger, R. L., and Llewellyn, E. J.: Atomic oxygen densities retrieved from Optical Spectrograph and Infrared Imaging System observations of O2 A-band airglow emission in the mesosphere and lower thermosphere, Journal of Geophysical Research: Atmospheres, 116, https://doi.org/10.1029/2010JD014640, 2011.

Su, S.-Y., Liu, C. H., Ho, H. H., and Chao, C. K.: Distribution characteristics of topside ionospheric density irregularities: Equatorial versus midlatitude regions, Journal of Geophysical Research: Space Physics, 111, https://doi.org/10.1029/2005JA011330, 2006.

Toledo-Redondo, S., Parrot, M., and Salinas, A.: Variation of the first cut-off frequency of the Earth-ionosphere waveguide observed by DEMETER, Journal of Geophysical Research: Space Physics, 117, https://doi.org/10.1029/2011JA017400, 2012.

Xu, W. and Marshall, R. A.: Characteristics of Energetic Electron Precipitation Estimated from Simulated Bremsstrahlung X-ray Distributions, Journal of Geophysical Research-Space Physics, 124, 2831-2843, 10.1029/2018ja026273, 2019.

Xu, W., Marshall, R. A., Tyssøy, H. N., and Fang, X.: A Generalized Method for Calculating Atmospheric Ionization by Energetic Electron Precipitation, Journal of Geophysical Research: Space Physics, 125, e2020JA028482, https://doi.org/10.1029/2020JA028482, 2020.

Yu, X., Yuan, Z., Yu, J., Wang, D., Deng, D., and Funsten, H. O.: Diffuse auroral precipitation driven by lower-band chorus second harmonics, Nature Communications, 14, 438, 10.1038/s41467-023-36095-x, 2023.

Zhao, S., Liao, L., Shen, X., and Lu, H.: Solar Cycle Variation of Radiated Electric Field and Ionospheric Reflection Height Over NWC Transmitter During 2005–2009: DEMETER Spacecraft Observations and Simulations, Journal of Geophysical Research: Space Physics, 129, e2023JA032282, https://doi.org/10.1029/2023JA032282, 2024.

---

## Author Comment (AC2)

Response to Reviewer Comment

We sincerely appreciate the reviewer's thorough assessment of our manuscript and their constructive comments. Below we provide a detailed response to each of the major and minor comments.

Major Comment 1: Reflection Height Assessment
The assessment of reflection height is improved from the previous version of this paper, but it still appears to be incorrect. The measurement of the first cutoff frequency altitude, as shown in Figure 1, is sufficient. But the model-derived reflection height (section 2.4) is problematic. The model provides an electron density (as in Figure 2); the authors then fit a Wait and Spies profile (equation 2) to the model D-region profile. They extract h', the Wait parameter, and use that as the reflection height. But, as I noted in the GRL review, h' is not the reflection height; it is merely a reference height for the electron density profile. The reflection height is correctly described in the paper as the height where $X = Z$ in the Ratcliffe formulation, or $w_p^2 = w*nu$ (equation 3, except a factor of $2*pi$ is missing).
The next steps in the paper are confusing. The authors use the first waveguide cutoff frequency from the ELF data, plug that into equation 3, and then solve for h'... But they already assumed a reflection height from the cutoff frequency through $f = c/h'$ for the cutoff frequency. This circular logic is confusing, and it is not clear (and not stated) which h' is used in the rest of the paper.
The assessment of the reflection height from the model can be made much simpler. Use the electron density from the model to calculate $w_p^2$. Use equation 4 to calculate nu, the collision frequency profile. Now, you can plot $X/Z$ as a function of altitude (on the y-axis) and frequency (on the x-axis), just as Ratcliffe did in his book. For the frequency of interest (say, the f1 determined from the ELF data), you can find the altitude where that frequency reflects.
The authors should still point out that for ELF/VLF frequencies, this is not a hard reflection; it occurs over a range of 5-10 km in altitude. But it is true that a lower effective reflection height is consistent with precipitation, as shown in Marshall and Cully [2020].
We thank the reviewer for this suggestion. We have calculated the reflection height exactly following your suggestion. We first used equation 4 to calculate the collision frequency $\nu$ at different heights z, and the plasma frequency $f_p$ at those heights was calculated using the model's electron density. These two parameters were then substituted into the equation $f_p^2 = f_1\nu$, leading to the relationship between different wave frequencies and their reflection heights (equation 6). We then used the observed $f_1$ from the EFD payload and equation 6 in order to obtain the relation between the reflection height h′ and the frequency $f_1$.

We admit that there was a logical issue in our description, which likely caused confusion regarding the relationship between z and h′. To address this, we have revised section 2.4, providing a clearer distinction between z and h′. Specifically, in line 213, we describe: "when discussing specific reflection heights at a particular frequency, h' will be used". Furthermore, in line 217, we clarify equation 6: "we can obtain the relationship between different wave frequencies and their reflection height." In lines 220-223, we offer a detailed explanation of how the data is obtained and how the reflection height is solved: "In this formulation, Ne

(z) represents the electron density (in $cm^{-3}$) at reflection height z, obtained from PyGPI5 simulations. The cutoff frequency $f_l$ corresponds to the wave frequency that is reflected at a specific height z. Using equation 6, h′ can be determined by finding the height z at which the calculated cutoff frequency matches the observed $f_l$ from CSES EFD measurements. "

These revisions should clarify the logic and resolve any confusion between z and h′. We greatly appreciate the reviewer's insight, and we believe these changes will enhance the clarity of the manuscript.

We also appreciate the reviewer's suggestion to clarify the nature of ELF/VLF reflections. Following this advice, we have added the following statement in lines 223-224: "For ELF/VLF frequencies, these waves are not reflected at a single altitude, but more likely over a range of 5-10 km around the reflection altitude. However, as pointed out by Marshall and Cully (2020), a reflection height that is lower than typical values was found to be more consistent with energetic electron precipitation." This addition strengthens our interpretation of the observed decrease in reflection height during precipitation events.

**Major Comments:**
2. X-ray observations: This comment is repeated from my GRL review, since I do not see that it was addressed. X-ray observations are shown, which give a spatial distribution similar to the particle fluxes. But apart from a spatial distribution, and a general amplitude correlation (not shown; a figure would be nice). Furthermore, the energy range of the X-ray observations is concerning. These energies (0.9--35 keV) are going to be dominated by bremsstrahlung from auroral electrons (<50 keV), NOT the relativistic electrons usually associated with EPP. Yet, it is intriguing that the X-ray fluxes seem to peak around L=5, clearly sub-auroral and in the radiation belts. Did the authors filter the data to remove times of auroral precipitation somehow? If so, this should be explained. Then, what were the fluxes of lower-energy precipitation electrons at this time? If lower-energy precipitation (<50 keV) is present, at high fluxes as is typical, then the x-rays may have been produced by those electrons, are we would not expect them to be strongly correlated with the electron fluxes or the reflection height (since those electrons deposit energy above the ELF/VLF reflection heights). But all of this information is missing, so cannot be properly assessed.

Thank you for the helpful comments and suggestions. Our recent paper (Liao et al., 2025) shows that areas with higher nightside bremsstrahlung X-ray rates are found in the outer radiation belt (L>4), the South Atlantic anomaly zone, and the slot region (2<L<3). Additionally, strong X-ray radiation can be observed in regions with an L value of approximately 1.7 and a geographic longitude of 150 to 200 degrees. The areas with high X-ray rates detected by CSES correspond closely to the areas with high-energy particle fluxes observed by CSES satellites (Figure 1 (b, c)). The bremsstrahlung process is extremely inefficient for <50 keV electrons, and it becomes more important for MeV electrons (Berger and Seltzer, 1972; Lehtinen et al., 1999). Moreover, the X-rays generated by hundred-keV to MeV electrons are not necessarily in the same energy range, but mostly in the keV range (Lehtinen et al., 1999; Berger and Seltzer, 1972; Xu and Marshall, 2019). As such, the 0.9-35 keV X-rays measured by CSES are not

necessarily produced by auroral electrons since the bremsstrahlung process for <50 keV electrons is inefficient, but more likely produced by hundred-keV and MeV electrons. Due to the limited energy resolution of CSES, we emphasize that X-rays with energies higher than 35 keV could be recorded by CSES, but cannot properly registered.

This correlation is supported by the simulation conducted by one coauthor of this manuscript (Xu and Marshall, 2019), which indicated that the X-rays detected at satellite altitudes primarily originate from the bremsstrahlung production source region. Although bremsstrahlung process becomes more efficient when the electron energy increase, however the distribution of X-ray rates aligns more closely with the electron flux distribution in the energy range of 100-800 keV (Figure 1 (b)) compared with the electron flux distribution in the energy range of 800 keV-3 MeV (Figure 1 (c)). This conclusion has also been confirmed by the spatial cross-correlation analysis by the calculation of Bivariate Moran's I. Unlike the Pearson or Spearman correlation coefficient, the Bivariate Moran's I consider the effect of spatial location on the correlation. The correlation between X-ray rate and electron flux is strong at 100-800 keV (0.77, p-value < 0.01) and moderate at 800 keV to 3 MeV (0.62, p-value < 0.01). That is because electrons with energies below 800 keV tend to have more precipitation energy backscattered into space, which can be detected by satellites. In contrast, the bremsstrahlung photons produced by MeV electrons are more likely to be emitted forward since they are directed downward into the atmosphere (Xu and Marshall, 2019b). X-rays in the radiation belt, particularly in the range of 2 < L < 3 from 150° E to 260° E in the northern hemisphere, maybe from lightning-induced electron precipitation (LEP). Another apparent feature is, in the energetic electron flux distribution map for 800 keV to 3 MeV, there is a notable slot region between 2 < L < 3 in the longitudinal range of 300° E to 360° E (Figure 1 c). This slot region does not appear in the distribution maps for the 100-800 keV range (Figure 1b) since the specific slot region is highly energy dependent (e.g., Voss et al., 1984; Linzmayer et al., 2024).

[Figure]

Figure 1.(a) The global nighttime X-ray distributions from 2019 to 2021. (b) The global nighttime electron flux distributions for electrons in the energy range of 100–800 keV from 2019 to 2021. (c) The same format as (b) but for the electrons in the energy range of 800 keV– 3 MeV.

From the above results, it can be seen that the night-side X-rays detected by the CSES satellite in the slot region and SAA regions have a low correlation with auroral electrons, which are mainly radiated due to the ionization of the neutral component by electrons in the 100-800 keV range. In the outer radiation belts, as stated by the reviewer, although we currently have no observations that can rule out the influence of auroral electrons (the detection range of CSES satellite electrons in the low-energy band of energy is 100 keV- 3 MeV), the precipitated electrons in the range of 100-800 keV should also play a very important role in the resulting X-ray.

[Figure]

Figure 2. X-ray profiles under different conditions. The figure shows the X-ray distribution in the background condition (red line) and during precipitation condition (blue line), using the average values measured by the CSES-HEPP-X in 2019.

According to the simulation study by Xu et al. (2019), electrons with energies exceeding 100 keV can generate X-rays with energies greater than 10 keV. We conducted a zonal analysis of the CSES satellite data, dividing the observation areas into two categories: high-latitude regions affected by radiation belt electron precipitation and low-latitude regions unaffected by radiation belt electrons. Through an annual average X-ray analysis, we found that the differences between high and low latitude regions become more pronounced for X-rays with energies exceeding 20 keV (Figure 2). This distinction is likely caused by radiation belt electron precipitation. However, determining whether these X-rays are produced by auroral electrons or higher-energy electrons will require further in-depth research in future work. In future work, we will also consider the reviewer's suggestion to find suitable observations of auroral electrons that have eliminated its effect.

**Minor comments:**
Figure 2: These plots should be extended down to 40 km or so to show where the new ionization profile blends back into the background profile.
Also, it's not clear why the authors show both 1500 keV and 1 MeV, and in that order. Those energies seem too close to show any meaningful difference. Why not 100 keV, 300 keV, and 1 MeV?
Your suggestion is very helpful and has improved the clarity of our manuscript. We have implemented the following changes to Figures 2 and 3:

- Extended the altitude range down to 40 km to better illustrate where the new ionization profiles merge with the background profile.
- Replaced the previously similar energy values (1500 keV and 1 MeV) with a more informative set: 100 keV, 300 keV, and 1 MeV. This selection provides a clearer demonstration of the energy-dependent variations in electron penetration depths.

The results in Table 1 are a nice addition to the paper, but the authors should explain what "Moran's I" means to those who are unfamiliar with that method.

Moran's I is a spatial autocorrelation statistical method used to measure the degree of clustering or dispersion of data in space. We chose this method because it effectively quantifies the correlation between X rays, precipitated electrons and reflection height in the spatial distribution.
The values of Moran's I typically range from -1 to +1:
- Positive values (close to +1) indicate positive correlation.
- Negative values (close to -1) indicate    negative correlation.
- Values close to zero indicate no apparent correlation.
We have revised Lines 275-276: "which is a spatial correlation statistical method and helpful in determining the correlation of different variables in the space. "

Reference:

Berger, M. J. and Seltzer, S. M.: Bremsstrahlung in the atmosphere, Journal of Atmospheric and Terrestrial Physics, 34, 85-108, https://doi.org/10.1016/0021-9169(72)90006-2, 1972.

Lehtinen, N. G., Bell, T. F., and Inan, U. S.: Monte Carlo simulation of runaway MeV electron breakdown with application to red sprites and terrestrial gamma ray flashes, Journal of Geophysical Research-Space Physics, 104, 24699-24712, 10.1029/1999ja900335, 1999.

Liao, L., Zhao, S., Li, Q., Dong, X., Lu, H., and Shen, X.: CSES Satellite Observation of Nightside Bremsstrahlung X-Ray Distributions Induced by Energetic Electron Precipitation, Journal of Geophysical Research: Space Physics, 130, e2024JA033700, https://doi.org/10.1029/2024JA033700, 2025.

Linzmayer, V., Němec, F., Santolík, O., and Kolmašová, I.: Lightning-Induced Energetic Electron Precipitation Observed in Long-Term DEMETER Spacecraft Measurements, Journal of Geophysical Research: Space Physics, 129, e2024JA032713, https://doi.org/10.1029/2024JA032713, 2024.

Voss, H. D., Imhof, W. L., Walt, M., Mobilia, J., Gaines, E. E., Reagan, J. B., Inan, U. S., Helliwell, R. A., Carpenter, D. L., Katsufrakis, J. P., and Chang, H. C.: Lightning-induced electron precipitation, Nature, 312, 740-742, 10.1038/312740a0, 1984.

Xu, W. and Marshall, R. A.: Characteristics of Energetic Electron Precipitation Estimated from Simulated Bremsstrahlung X-ray Distributions, Journal of Geophysical Research: Space Physics, 124, 2831-2843, https://doi.org/10.1029/2018JA026273, 2019.

---

## Author Response (AR2)

Response to Reviewer Comment

We sincerely thank reviewer for the thorough and constructive review. The detailed feedback and technical suggestions have greatly improved the clarity, accuracy, and presentation of our manuscript. We have carefully addressed all eight points, including clarifying the averaging window size, refining methodological descriptions, correcting terminology, and enhancing figure readability. The reviewer's careful attention to detail has strengthened the rigor and transparency of our study. We believe these revisions significantly improve the manuscript's quality and reader comprehension. Below is our detailed point-by-point response and description of the changes made.

1. Line 173: There is still no information about the size of the averaging windows. This is an important information for the reader.
We have added the following sentence in Lines 173-174 :"Both smoothing steps use a window size of 45 data points."

2. Line 214: "A commonly used model for the collision frequency $v$ at reflection height z is (Gołkowski et al., 2018)". It shows the collision frequency as a function of altitude, not at the reflection height.
We thank the reviewer for this important terminological correction. The reviewer is absolutely correct that the equation describes how collision frequency varies with altitude rather than specifying the collision frequency at a particular reflection height. "A commonly used model for the collision frequency ν as a function of altitude z is (Gołkowski et al., 2018) :" We have revised the text in Line 216.

3. Line 223-225: z and reflection height are mixed up in the text. "In this formulation, $Ne(z)$ represents the electron density (in cm⁻³) at reflection height z, obtained from PyGPI5 simulations." Ne(z) only represents the vertical density profile. In general, this section remains quite confusing. Did I understand the following right?
Inserting equation (4) and (5) into equation (3) + using the Ne(z) from PyGPI5 simulations and the cutoff frequency from the ELF data (derived in section 2.3)
obtains equation (6), which can be solved for z. Where the computed z is then the reflection height for the cutoff frequency f1,- Please rephrase the description of method 2 for better legibility.
We sincerely thank the reviewer for identifying this confusion and providing a more clear interpretation of the methodology. The reviewer's understanding is absolutely correct, and we acknowledge that our original description was poorly organized and contained terminological inconsistencies.

"By substituting Eq.(4) and Eq.(5) into Eq.(3), we derive the condition that must be satisfied among the wave frequency, height, and electron density for electromagnetic wave reflection to occur in the D region." We have revised the text in Lines 221-222.

"In this formulation, $Ne(z)$ represents the vertical electron density profile (in cm⁻³) obtained from PyGPI5 simulations. It describes how the electron density varies with altitude, rather than providing a single value at a specific height. To determine the reflection height corresponding

to a specific wave frequency $f_1$, we substitute the observed cutoff frequency $f_1$ (derived from CSES EFD data as described in Section 2.3) into equation (6). We then perform a numerical iteration over different altitudes z until the calculated cutoff frequency from equation (6) matches the observed $f_1$. The altitude z that satisfies this condition is defined as the effective reflection height h' for the given frequency." We have revised the text in Lines 224-229.

4. In section 2.3, the cutoff frequency is determined by ELF measurements. It is also stated in section 2.3 that f1 = c/(2h). You could easily compute h from this. Why do you apply the other two methods?

Thank you for the valuable comment. Indeed, the formula $f_1 = c/2h'$ in Section 2.3 can directly calculate the reflection height from the cutoff frequency. Our study follows a stepwise scientific approach:

Section 2.3 mainly presents observational results, where reflection heights are derived from CSES ELF data using this formula, revealing a decrease in reflection height in regions affected by energetic particle precipitation. This is an observational phenomenon requiring further physical explanation.

Section 2.4 uses the PyGPI5 model to simulate electron density profiles and applies two different methods to calculate reflection heights to verify whether the observed decrease is caused by energetic electron precipitation (EEP):
1.  Method one uses WS formula fitting, commonly applied in VLF inversion studies, providing results compatible with practical applications.
2.  Method two is based on the Ratcliffe relation, offering an analytical perspective on the reflection mechanism from plasma physics.

These two methods complement each other. When consistent results are obtained, it strengthens the reliability of the analysis and confirms that the observed reflection height variation is indeed caused by EEP rather than other factors. This multi-method validation elevates a simple observational correlation to a physically verified mechanism.

5. Lines 226-228: "For ELF/VLF frequencies, these waves are not reflected at a single altitude, but more likely over a range of 5- 10 km around the reflection altitude. However, as pointed out by Marshall and Cully (2020), a reflection height that is lower than typical values was found to be more consistent with energetic electron precipitation." These sentences are out of context. The information about the reflection height could have been raised earlier. And the information about the EPP could be placed in the discussion and why "However"?.
Thank you very much for your valuable comments. We have revised the manuscript accordingly to improve the clarity and logical flow:
We added the following sentence at Lines 155–157 to introduce the concept of ELF/VLF wave reflection over a range of altitudes earlier in the text: "It should be noted that ELF/VLF waves do not reflect at a single fixed altitude, but rather over a range of about 5–10 km around the

reflection height, which is important for explaining differences between observations and simulations."

The discussion related to energetic electron precipitation (EPP) and its connection to a lower reflection height has been moved to Lines 310–312 within the discussion section for better contextual relevance, with the following addition: "As pointed out by Marshall and Cully (2020), a reflection height that is lower than typical values was found to be more consistent with energetic electron precipitation, which supports our observed correlations."

6. Line 278-281: It is ok if the Moran method is not explained in detail. However, a reference is then needed where the method is described in detail. The meaning of Moran's I in table 1 needs to be explained in the text. What information is provided by Moran's I (e.g., Positive/negative correlation-> cumulation, dispersion, information from the value of Moran score)?

We thank the reviewer for this important clarification regarding the Moran's I analysis. We agree that additional explanation and proper referencing are needed to help readers interpret Table 1 results.

We have included Moran (1950) as the reference in Line 281. We also have expanded the text following Lines 281-284 to explain that Moran's I quantifies spatial autocorrelation, where: " Moran's I is a spatial autocorrelation statistic that measures the degree to which similar values cluster together in space. The statistic ranges from -1 to +1, where positive values indicate positive spatial correlation, negative values suggest negative spatial correlation, and values near zero indicate random spatial distribution."

7. During the first revision, I did not notice that the color code in Figures 4e and f was reversed. For better readability and consistency, it would be very helpful to arrange the color bar as in Figures 4a-d.

We thank the reviewer for the careful attention to the color coding details and the valuable suggestion. We understand the importance of color consistency for figure interpretation.

The physical process illustrated in Figure 4 is as follows: an increase in electron flux leads to a rise in X-ray rate, which subsequently causes a decrease in reflection height. To accurately reflect this physical relationship, we intentionally reversed the color bars in Figures 4e and 4f compared to Figures 4a–d. In this way, "high reflection height" is represented by cool colors, while "low reflection height" is represented by warm colors. This ensures that regions with high electron flux and high X-ray rate (both shown with warm colors) visually correspond to regions with low reflection height (also shown with warm colors), helping readers intuitively grasp the relationship among these physical quantities. Therefore, although the color bar direction in panels e and f differs from that in panels a–d, this design choice maintains consistency in physical interpretation. We believe this arrangement enhances the overall clarity of Figure 4. We appreciate the reviewer's detailed review and understanding.

8. Line 285: The meaning of the sentence is not clear. It also seems out of context. Are you

referring to table 1 or a figure? Isn't ITC more southwards (max. 20/30°N)? I assume 40-50°N is too high for the ITC.

Thank you very much for your careful reading and valuable comments. We agree that the original wording was inaccurate. We have revised the sentence to improve clarity and geographical correctness by changing "lies within" to "is near" regarding the Inter-Tropical Convergence Zone in Line 291: "the underlying reason is that this area is near the Inter-Tropical Convergence Zone."

References:

Gołkowski, M., Sarker, S. R., Renick, C., Moore, R. C., Cohen, M. B., Kułak, A., Młynarczyk, J., and Kubisz, J.: Ionospheric D Region Remote Sensing Using ELF Sferic Group Velocity, Geophysical Research Letters, 45, 12,739-712,748, https://doi.org/10.1029/2018GL080108, 2018.

Marshall, R. A. and Cully, C. M.: Chapter 7 - Atmospheric effects and signatures of high-energy electron precipitation, in: The Dynamic Loss of Earth's Radiation Belts, edited by: Jaynes, A. N., and Usanova, M. E., Elsevier, 199-255, https://doi.org/10.1016/B978-0-12-813371-2.00007-X, 2020.

Moran, P. A. P.: Notes on Continuous Stochastic Phenomena, Biometrika, 37, 17-23, 10.2307/2332142, 1950.